

# GaitTriViT and GaitVViT: Transformer-based methods emphasizing spatial or temporal aspects in gait recognition

Hongyun Sheng

School of Computing Science, University of Glasgow, Glasgow, United Kingdom

## ABSTRACT

In image recognition tasks, subjects with long distances and low resolution remain a challenge, whereas gait recognition, identifying subjects by walking patterns, is considered one of the most promising biometric technologies due to its stability and efficiency. Previous gait recognition methods mostly focused on constructing a sophisticated model structure for better model performance during evaluation. Moreover, these methods are primarily based on traditional convolutional neural networks (CNNs) due to the dominance of CNNs in computer vision. However, since the alternative form of Transformer, named Vision Transformers (ViTs), has been introduced into the computer vision field, the ViTs have gained strong attention for its outstanding performance in various tasks. Thus, unlike previous methods, this project introduces two Transformer-based methods: a completely ViTs-based method GaitTriViT, and a Video Vision Transformer (Video ViT) based method GaitVViT. The GaitTriViT leverages the ViTs to gain more fine-grained spatial features, while GaitVViT enhances the capacity of temporal extraction. This work evaluates their performances and the results show the still-existing gaps and several encouraging outperforms compared with current state-of-the-art (SOTA), demonstrating the difficulties and challenges these Transformer-based methods will encounter continuously. However, the future of Vision Transformers in gait recognition is still promising.

## INTRODUCTION

Gait is defined as the physical and behavioral biological characteristics exhibited by a human when walking upright, and it can be used to describe an individual's walking pattern. Gait recognition is the technology that identifies individuals based on their distinct walking patterns. Whilst other biometric features, such as faces, fingerprints and irises, can be used for identification purposes, the superiority of gait lies in its ability to be easily captured from a distance and the fact that it can be carried out without any subject cooperation or contact for data acquisition (*Nixon & Carter, 2006*). This makes gait recognition highly promising in real-world applications.

Gait recognition is an attractive method of identification. For instance, many video surveillance systems are only capable of capturing low-resolution video in suboptimal

Corresponding author
Hongyun Sheng,
hongyunsheng0421@gmail.com

lighting conditions. In the case of identifying bank robbers, they may wear masks to conceal their faces, gloves to prevent the capture of fingerprints, and hats to conceal hair and DNA. However, they invariably need to walk or run, which can be easily captured by gait analysis. In such scenarios, gait recognition emerges as a pivotal method for automatic identification (*Makihara, Nixon & Yagi, 2021*).

Research in the field of gait recognition is currently undergoing a transition from the evaluation stage to the application stage, with the potential for utilization in a variety of contexts, including forensics, social security, immigration control, and video surveillance. In several criminal cases, gait recognition has been adopted as evidence for conviction, underscoring its significance in legal proceedings. A notable example can be found in a 2011 forensic study that utilized gait features to provide evidence for identification (*Bouchrika et al., 2011*). Furthermore, there is a prevailing sentiment within the judicial system that gait analysis holds considerable potential as a valuable investigative tool (*Larsen, Simonsen & Lynnerup, 2008*). In Japan, a gait verification system for criminal investigation has been developed and is currently undergoing a trial phase by the National Research Institute of Police Science (*Iwama et al., 2013*). The biometric tunnel proposed by *Seely et al. (2008)* led to the first live demonstration of gait as a biometric and may still be the most promising future route of gait recognition in deployment, such as access control. The first commercial software for gait recognition 'Watrix' was released in Oct. 2018 and was developed by the Institute of Automation, Chinese Academy of Sciences (CASIA). The software can accept two videos from users, one of which is used as a gallery and the other as a probe. The software then produces a report of the match result.

The primary objective of this undertaking is the acquisition of authentic, effective and distinctive representations from target data, yet the endeavor faces numerous challenges in practical applications for a multitude of reasons. These include self-occlusion, viewing angles, walking status and carrying conditions such as the lugging of a bag (*Fan et al., 2023*; *Sepas-Moghaddam & Etemad, 2022*; *Wan, Wang & Phoha, 2018*). As a task with extensive application prospects, these challenges urgently need to be addressed.

A plethora of gait recognition methods have been developed, including Gait Energy Image (GEI) by *Han & Bhanu (2005)*, GaitSet, GaitPart, and GaitGL (*Chao et al., 2018*; *Fan et al., 2020*; *Han & Bhanu, 2005*; *Lin et al., 2022*). These methods have been shown to enhance the performance of traditional convolutional neural network (CNN) and recurrent neural network (RNN) architectures (*Fan et al., 2023*; *Sepas-Moghaddam & Etemad, 2022*). They utilize more sophisticated structures and deeper neural network layers to achieve improved performance in extracting representative features. This approach is particularly popular given the predominance of CNN-based methods in the domain of computer vision, where they have achieved notable success in image and video tasks that were previously unattainable for deep neural networks (*Dosovitskiy et al., 2020*).

However, the introduction of Vision Transformer (ViT) methods by *Dosovitskiy et al. (2020)* has recently resulted in remarkable advancements in a range of tasks, including object detection, image segmentation, and image classification (*Carion et al., 2020*;

*Chen et al., 2021*; *Dosovitskiy et al., 2020*; *Hong et al., 2022*), and researchers are continually enhancing the performance, proposing many advanced novel architectures, *e.g.*, Swin Transformer by *Liu et al. (2021)* and VideoMAE by *Tong et al. (2022)*, to endow ViT with more capabilities and potential. The multi-head attention mechanism, a distinctive feature of ViT, enables the acquisition of intricate spatial features at the frame level in video-based recognition tasks, a capability often compromised by down-sampling operations in CNN-based methods (*Alsehaim & Breckon, 2022*).

The integration of patch division and multi-head attention in ViT not only preserves the capacity to extract features from small regions but also facilitates the capture of long-range dependencies, a capability that is essential for gait recognition tasks that emphasize the simultaneous consideration of both local and global features (*Hou et al., 2022*). Furthermore, the adoption of a patch-based approach, where frames are regarded as patches subject to changes in scale, enables the fundamental ViT structure to attain sequence-level temporal attention. This sequence-level temporal attention, exemplified by the Video Vision Transformer (VViT), has been shown to be advantageous for gait recognition tasks (*Arnab et al., 2021*; *Liu et al., 2021*; *Neimark et al., 2021*). This article thus presents two customized methods that leverage Vision Transformer technology to address the gait recognition task.

The motivation behind this study stems from the desire to examine the impact of incorporating ViT into conventional gait recognition tasks and to achieve superior performance in comparison to state-of-the-art methods. To this end, the following works were undertaken.

This article introduces two Transformer-based gait recognition models: GaitTriViT and GaitVViT. GaitTriViT consists of a backbone for frame-level feature extraction, followed by two parallel Transformer-based branches. The local part spatial branch is designed for the extraction of fine-grained set-level features in local regions, while the global temporal branch is built for the extraction and aggregation of global features with temporal attention (*Fu et al., 2019*; *Rao et al., 2018*; *Zhang et al., 2020*). The final part of the model is multiple heads for classification, and then a fusion loss function is used to optimize the model. The technology employed includes the Vision Transformer (*Dosovitskiy et al., 2020*) and temporal clip shift and shuffle (TCSS) by *Alsehaim & Breckon (2022)*. GaitVViT adopts the technology from GaitGL (*Lin et al., 2022*) to construct the backbone structure, and the backbone is connected to a Video Vision Transformer Network to build the final structure (*Arnab et al., 2021*; *Neimark et al., 2021*). The Video Vision Transformer then models the features along the temporal dimension, generating the final features and predicted labels (*Arnab et al., 2021*; *Neimark et al., 2021*). The proposed methods are tested on two popular benchmarks: CASIA-B and OUMVLP (*Takemura et al., 2018*; *Yu, Tan & Tan, 2006*).

Portions of this text were previously published as part of a thesis (https://theses.gla.ac.uk/84475/1/2023ShengMSc%28R%29.pdf)

Overall, in this work, several contributions are made as shown below:

A customized Transformer-based method GaitTriViT leveraging three ViT blocks parallelly. The global and local features are both emphasized, along with the combination of spatio-temporal attention.

More than position embedding, the camera angle and walking status of subjects from different frame sequences are also emphasized and embedded, which are intended to enhance the robustness of gait recognition when facing challenges *e.g.*, cross-view and multiple walking status.

A customized method GaitVViT uses a Video Vision Transformer as an inherent aggregator regarding gait data as frames in order not set. Emphasizing the idea of sequence, the method is dedicated to enhancing the temporal modeling performance over the common framework.

The evaluation of proposed methods on two popular benchmarks and the comparison to state-of-the-art indicate the challenges and potential for a Transformer-based model in gait recognition tasks.

## RELATED WORKS

In recent research, gait recognition methods can be broadly categorized into two main classes: model-based and appearance-based (*Fan et al., 2023*, *2022*; *Hou et al., 2022*; *Santos et al., 2023*). Model-based methods estimate the underlying human body structures from raw data and use them as input *e.g.*, 2D/3D poses and the SMPL model (*Cao et al., 2016*; *Liao et al., 2020*; *Loper et al., 2015*; *Martinez et al., 2017*; *Roy, Sural & Mukherjee, 2012*). For example, GaitPT by *Catruna, Cosma & Radoi (2024)* and GaitFormer by *Cosma & Radoi (2022)* leverage pose estimation skeletons to capture unique walking patterns through transformer architecture, achieving remarkable outcome. In contrast, appearance-based methods favor directly extracting feature representations of human walking patterns from gait silhouettes. Due to the challenges of gait recognition tasks, which often involve long distances and low resolutions (*Nixon & Carter, 2006*), recent studies have emphasized the practicality of appearance-based methods for their robustness (*Fan et al., 2022*).

Person re-identification (Re-ID) tasks work similarly to gait recognition. In Re-ID tasks, when being presented with a person-of-interest (query), the method tells whether this person has been observed in another place (time) by another camera. From the perspective of computer vision, the most challenging problem in Re-ID is how to correctly match two images of the same person under intensive appearance changes, such as lighting, pose, and viewpoint (*Zheng, Yang & Hauptmann, 2016*). In other words, Gait Recognition can be regarded as a subset of person Re-ID leveraging gait as input. There are many Transformer-based methods have been proposed, *e.g.*, VID-Trans-ReID by *Alsehaim & Breckon (2022)* and TransReID by *He et al. (2021)*. The shuffle operation on the feature map and the choice of loss functions introduced in these methods also served as inspiration in this article.

Among the appearance-based gait recognition approaches, the way of feature extraction can be discussed from three perspectives: Spatial, Temporal, and Transformer.

### Spatial feature extraction

In gait recognition research, the introduction of deep convolutional neural networks was pioneered by *Wu et al. (2017)*, they studied an approach to gait-based human identification

*via* similarity learning by deep CNNs, aiming to recognize the most discriminative changes of gait patterns which suggest the change of human identity with a pretty small group of labeled multi-view human walking videos.

Subsequently, GaitSet, proposed by *Chao et al. (2018)* adopted the strategy of dividing feature maps into strips from prior person re-identification research, enhancing the description of the human body. It has been adopted by many following researchers ever since; GaitPart introduced by *Fan et al. (2020)* pushes forward the part-based concept further, presenting a part-dependent approach, they argued that the part-based schemas applied in gait recognition should be part-dependent rather than part-independent, because despite there are significant differences among human body parts in terms of appearance and moving patterns in the gait cycle, it is highly possible that different parts of human body share the common attributes, *e.g.*, color and texture. Thus, the parameters are designed part-dependent in FConv (focal convolution) layers to generate the fine-grained spatio-temporal representations; GaitGL developed by *Lin, Zhang & Yu (2020)*, *Lin et al. (2022)* elaborated the disadvantage of extraction from either global appearances or local regions of humans only. They argued the representations based on global information often neglect the details of the gait frame, while local region-based descriptors cannot capture the relations among neighboring regions, thus reducing their discriminativeness. Thus, they effectively combined global visual features and local region details, demonstrating the necessity to address both aspects simultaneously; SMPLGait by *Zheng et al. (2022)* aims to explore dense 3D representations for gait recognition in the wild. Leveraged the human body mesh to acquire three-dimensional geometric information, they proposed a novel framework to explore the 3D skinned multi-person linear (SMPL) model of the human body for gait recognition; MetaGait designed by *Dou et al. (2023)* argued that there are still conflicts between the limited binary silhouette and numerous covariates with diverse scales. Their model can learn an omni-sample adaptive representation by injected meta-knowledge in a calibration network of the attention mechanism, which could guide the model to perceive sample-specific properties; also (*Fan et al., 2022*), in their code repository OpenGait, drew insights from previous state-of-the-art methods and introduced GaitBase, which achieved excellent results. These studies often stack deeper convolutional layers or complex architecture to capture fine-grained, more robust, and discriminative features, to meet the various challenges of gait recognition tasks.

## Temporal feature aggregation

The temporal modeling has consistently remained a significant focus in gait recognition tasks due to the inherent periodicity of walking patterns in the time dimension, *i.e.*, gaits are repeating loops. Presently, there are three popular directions in existing research: 3D convolutional neural network (3DCNN)-based, Set-based, and LSTM-based approaches.

Among 3DCNN-based methods, *Wolf, Babaee & Rigoll (2016)* and *Tran et al. (2015)* directly employ 3DCNN to extract spatio-temporal features from sequential data. They indicated that 3D Convolutional Networks are more suitable compared to 2D and a homogeneous architecture with small [3, 3, 3] convolution kernels in all layers is among the best performing architectures for 3D Convolutional Networks. However, this approach

often encounters training difficulties and yields suboptimal performance; set-based methods view frames within a cycle as an unordered set since humans can easily identify a subject from a shuffled gait sequence. Furthermore, due to the short duration of each gait cycle, long-range dependencies and duplicate gait cycles are considered redundant. Take GaitSet, for example *Chao et al. (2018)*. In contrast to prior gait recognition methods which utilize the frames of either a gait template or a gait sequence, they argued that the temporal information is hard to preserve in the template, while the sequence keeps extra unnecessary sequential constraints and thus has low flexibility. So, they present a novel perspective regarding gait as a set consisting of independent frames. Their method is immune to permutations of frames and can naturally integrate frames from different videos under different scenarios. These set-based methods typically study spatial features frame by frame and then perform temporal aggregation at the set level. On the other hand, LSTM-based methods like GaitNet by *Zhang et al. (2019)* argue that for each video frame, the current feature only contains the walking pose of the person in a specific instance, which can share similarity with another specific instance of a very different person. Therefore, modeling its change is critical. That is where temporal modeling architectures like the recurrent neural network or long short-term memory (LSTM) work best. They use a three-layer LSTM network to extract ordered sequence features. These LSTM-based methods are capable of capturing features between consecutive frames, often yielding slightly better performance. However, they lack efficiency and robustness to noise; therefore, many researchers still prefer set-based approaches.

## Attempts with transformer

Multiple works have tried to tackle the gait recognition task by introducing the Vision Transformers (ViTs) (*Dosovitskiy et al., 2020*). Since the ViTs are more compact in contrast to a multi-layer CNNs when they need to achieve similar performance, and ViTs are full of potential in the computer vision field. For example, Gait-ViT by *Mogan et al. (2022)* emphasized the lack of attention mechanism in Convolutional Neural Networks despite their good performance in image recognition tasks. The attention mechanism encodes information in the image patches, which facilitates the model to learn the substantial features in the specific regions. Thus, this work employs the Vision Transformer (ViT) integrated attention mechanism naturally. However, they used the gait energy image (GEI) to model the time dimension by averaging the images over the gait cycle; *Pinić, Suanj & Lenac (2022)* proposed a self-supervised learning (SSL) approach to pre-train the feature extractor, which is a Vision transformer architecture using the DINO model to automatically learn useful gait features (*Caron et al., 2021*; *Cui & Kang, 2022*) proposed GaitTransformer. They used a multiple-temporal-scale transformer (MTST), which consists of multiple transformer encoders with multi-scale position embedding, to model various long-term temporal information of the sequence. Furthermore, *e.g.*, *Yang et al. (2023)* and *Zhu et al. (2023)* also explore the vision transformer in gait recognition. However, the transformer-based methods have not outperformed CNN-based methods on the popular testing benchmarks and other challenging in-the-wild gait datasets (*Fan et al., 2023*).

# PROPOSED METHODS

## Introduction

Having studied the previous works, two transformer-based Gait Recognition methods are proposed: GaitTriViT and GaitVViT. GaitTriViT integrates the strengths of ViT and incorporates excellent ideas from previous works and recent advancements in related fields. It places emphasis on both global and local regions, considering both temporally and spatially. Furthermore, several modifications are also made in this work, which differs from the common gait recognition frameworks. GaitVViT enhances the temporal modeling ability of a common framework leveraging a Video Vision Transformer (VViT), which works as a novel temporal pooling (TP) module.

## Common framework

Recent studies indicate a common framework in various gait recognition tasks (*Fan et al., 2023*), as shown in Fig. 1. This framework abstracts complex structures into multiple modules, omitting internal details. The backbone maps the input gait sequence to features, typically used to extract frame-level spatial information. The TP module then aggregates feature maps along the time dimension, with operations *e.g.* max pooling, RNNs (*Fan et al., 2020*; *Iwama et al., 2013*; *Tran et al., 2021*).

Subsequently, the horizontal pooling (HP) module divides the feature map into several different parts in the horizontal direction, in line with the part-dependent concept introduced by *Fan et al. (2020)* and processes them independently. The Head may include several fully connected layers to obtain predicted labels, and it may also have a batch normalization neck (BNNeck) to map the features to different spaces before calculating the loss (*Luo et al., 2020*). Finally, both triplet loss and cross-entropy loss are used to optimize the model simultaneously (*De Boer et al., 2005*; *Hermans, Beyer & Leibe, 2017*; *Hoffer & Ailon, 2015*; *Rubinstein & Kroese, 2004*).

## GaitTriViT

### Pipeline

The common framework of gait recognition often apply the temporal pooling (TP) module and horizontal pooling (HP) module serially in model, which leads to the TP module may inevitably lose some low-rank features affecting the model performance, our method treats the original serial TP and HP modules as two separate and parallel branches, providing more potential to preserve refined patterns in different dimensions, as shown in Fig. 2. The overall structure can be divided into several modules, including the backbone, local part spatial branch, global temporal branch, BNNeck head, and optimizer (*Luo et al., 2020*). Aligned silhouettes are fed into the Transformer-based backbone first, after the feature extraction, the feature maps go separately into two parallel branches. The local part spatial branch works to extract the spatial features focused on different local parts, and the global temporal branch works to model the high-level spatio-temporal features. Those local and global features are delivered to classification heads with BNNeck to generate the discriminative final feature representations and predicted labels (*Luo et al., 2020*), which

**Figure 1 The common framework of gait recognition model including backbone, TP, HP, head and loss.** From left to right: Inputs are silhouette sequence; backbone network maps inputs to feature embeddings; TP stands for temporal pooling to aggregate temporal dimension; HP stands for horizontal pooling to treat feature maps as divided parts; the last part is classification head and loss function.

will be used to calculate the losses for model optimization. Equations (1), (2), (3), (4) below briefly describe the workflow of model GaitTriViT in Fig. 2.

$$F_{bone}^i = ViT_{bone}\left(\sum_{j=0}^{n}\left[emb_{pos}^j;\ emb_{case}^j;\ Emb^j(F^i)\right]\right) \tag{1}$$

$$F_{local}^{it} = Head_{local}\left(ViT_{local}\left(Seg^t\left(F_{bone}^i\right)\right)\right) \tag{2}$$

$$F_{global}^i = Head_{global}\left(Attention\left(ViT_{gobal}\left(F_{bone}^i\right)\right)\right) \tag{3}$$

$$Output_{GaitTriViT}^{F^i} = L\left(F_{global}^i, \sum_{t=1}^{parts} F_{local}^{it}\right) \tag{4}$$

where $Emb^j$ is the j-th embedding after patch cutting and flatten, along with position and case embeddings, $ViT_{bone}$ is the Backbone ViT block, $Seg^t$ is the t-th part of Patch Shuffle and Part Segment, $ViT_{local}$ represents the ViT block in local branch, $Head_{local}$ is Classification Head for local branch, similarly, $ViT_{global}$ and $Head_{global}$ are counter parts in global branch, here $Attention$ is the Temporal Global Attention layer, finally features pass through function $L$ to get final loss.

### Backbone

In this work, a Vision Transformer block is used to build the backbone to extract frame-level spatial features (*Dosovitskiy et al., 2020*). The original gait silhouette is in the form of a frame sequence $V_i = \{F_0,\ F_1, \ldots, F_t\}$, where each frame, after data rearrangement and pre-processing, is in the form of $F_j \in \mathbb{R}^{H \times W \times C}$, with $H$, $W$, and $C$ representing the height, width, and channels of the frame image, respectively. Each frame is divided into multiple patches of the same size as the original article does, *i.e.*, $F_j = \{P_0, P_1, \ldots, P_n\}$ (*Dosovitskiy et al., 2020*). However, drawing inspiration from works by *He et al. (2021)* and *Wang et al. (2022)*, this work also adopts their patch embedding strategy of allowing patches to overlap with each other. This approach helps the model to focus on local information while strengthening the connections between adjacent patches and reducing feature loss at the patch edges, which meets the need of this task to focus on body parts while not ignoring the constraints among each body part.

When cutting images into patches, different from the traditional method, the overlapping strategy is adopted for a more robust performance as Fig. 3 shows.

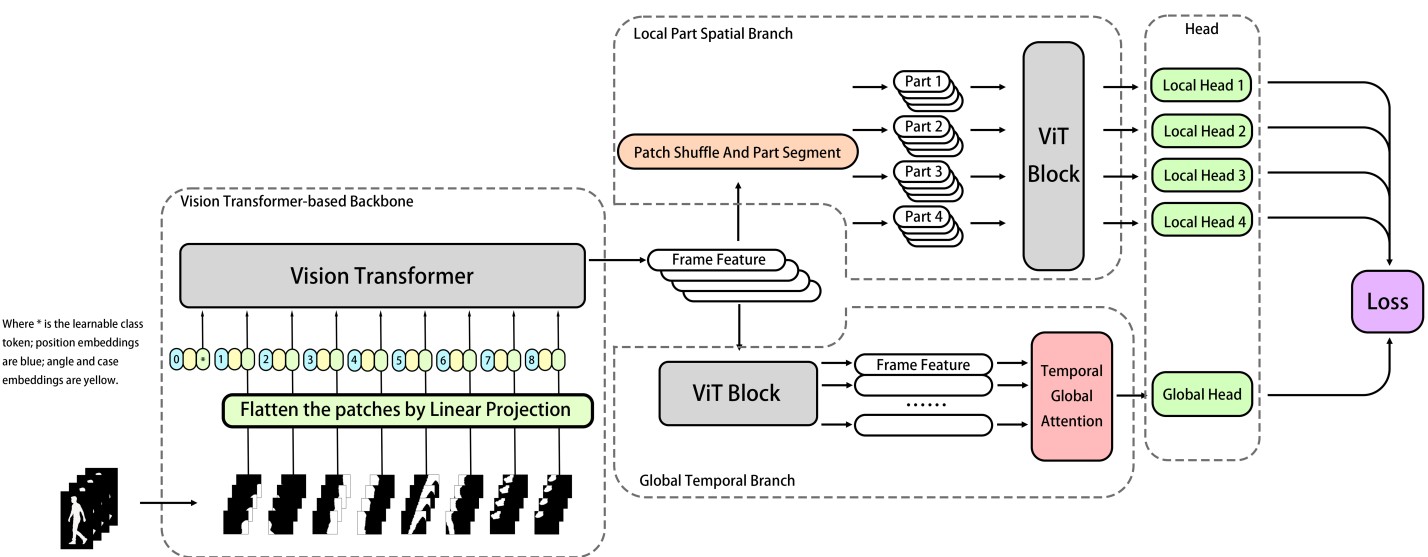

**Figure 2 The pipeline of GaitTriViT including ViT-based backbone, local-part-spatial branch, global-temporal-branch, heads and loss.** From left to right: Inputs are cut into patches and fed into ViT-based Backbone; the features then treated by two parallel branches, the upper one is Local-Part-Spatial Branch for detailed local feature extraction; the lower one is global-temporal-branch to obtain feature in frame-bundle-level; then multiple Heads will map them and send to Loss.

$$N = \frac{H + d - s}{s} \times \frac{W + d - s}{s}. \tag{5}$$

In Eq. (5), $N$ is the number of divided patches, $d$ is the patch size, and $s$ is the stride length. After the patches are generated, we need to flatten them into tensors of 1-D dimensions using linear projection $\ell$. Moreover, a learnable class token $P_{cls}^j$ is inserted at the head position to represent the overall features of this frame.

$$F_j = \left[ P_{cls}^j ; \ell\left(P_0^j\right); \ell\left(P_1^j\right); \ldots ; \ell\left(P_N^j\right) \right]. \tag{6}$$

$$E_j = F_j + \lambda_1 E_{pos} + \lambda_2 E_{angle} + \lambda_3 E_{case.} \tag{7}$$

Following the Vision Transformer original article (*Dosovitskiy et al., 2020*), A learnable position embedding $E_{pos} \in \mathbb{R}^{N+1 \times D}$ is added to represent the spatial position of each patch. Furthermore, due to the challenges posed by cross-view and different walking statuses in appearance-based gait recognition tasks, we manually incorporate information that represents different subject appearances and various camera angles into the patch embedding. Many studies have demonstrated the effectiveness of this operation, *e.g.*, research by *He et al. (2021)* and *Alsehaim & Breckon (2022)*, indicating that these lightweight learnable embeddings perform well tackling cross-view and cross-status tasks. For example, the current frame sequence is selected from a video where a subject is captured by a camera at the front while carrying a bag, which means the camera angle is 0 and the walking status is bag carrying. Similar to position embedding, we introduce case embedding $E_{case} \in \mathbb{R}^{\{c \times D\}}$ and angle embedding $E_{angle} \in \mathbb{R}^{a \times D}$, $c$ is the total number of existing walking situations, $a$ stands for the total number of different camera angles. Then,

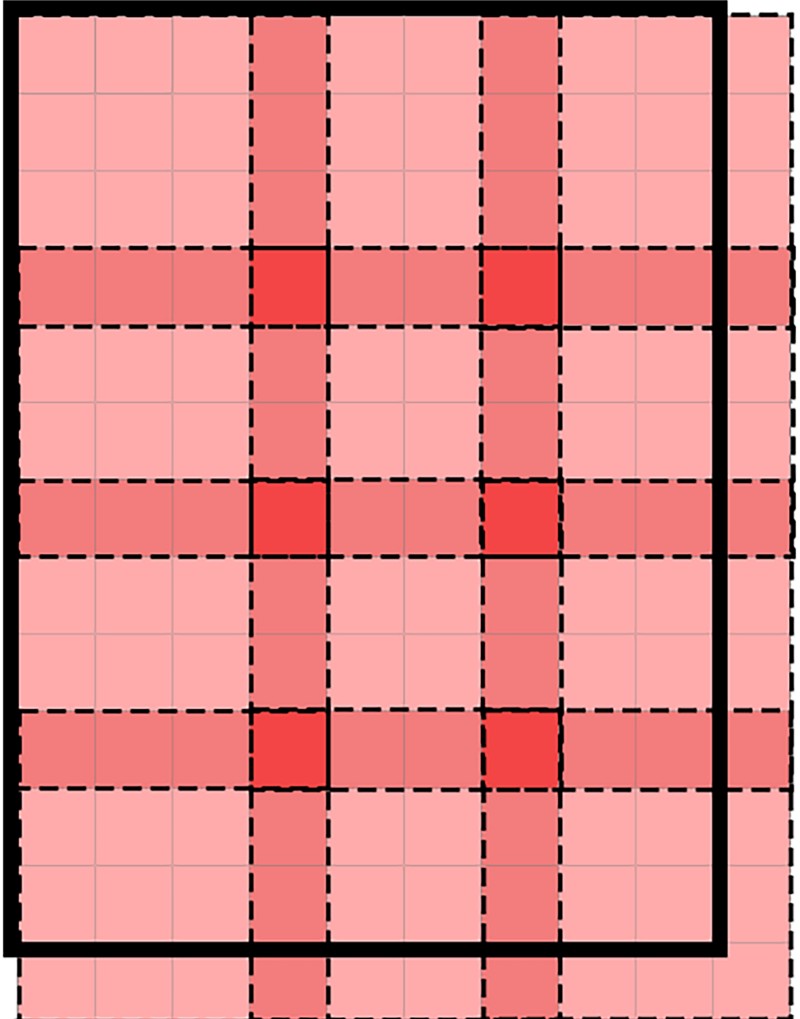

**Figure 3  The illustration of patches overlapping strategy in backbone of GaitTriViT.** The bold black bounding box indicates the full image; each dashed-line square represents a single patch, while the darker-colored areas correspond to overlapping patch edges.

we add these four altogether in proportions denoted by $\lambda_1$, $\lambda_2$, and $\lambda_3$, where we generate the final patch embedding $E_j$.

### Local part spatial branch

For each frame sequence representing a unique subject ID with a unique camera angle and walking status, only several frames are selected in one batch, which is regarded as a frame bundle. For each frame bundle $B$ that has undergone processing by the backbone, it now exists as follows:

$$B = \left[ F_0\left\{P_{cls}^0, P_0^0, \ldots, P_N^0\right\}; \ldots; F_T\left\{P_{cls}^T, P_0^T, \ldots, P_N^T\right\}\right]. \tag{8}$$

The bundle is sent to two branches, one of which is the local part spatial branch that will be discussed in this section. It corresponds to the HP module in the gait recognition

common framework and is used to extract spatial features at the frameset level within the bundle. *Chao et al. (2018)* and their proposed method GaitSet suggest that gait recognition does not require long-term dependencies and that treating gait frames as a set can improve model robustness. As our method has two separate branches to process the same feature representations in parallel, each branch can go further on its own specialized work and has no worries about affecting another branch. In the local part branch, the work is all about spatial local features. Therefore, we merge patches belonging to different frames but at the same position within the bundle as group $G_n$, where $n \in \{1, 2, \ldots, N\}$, these groups form a new bundle $\widehat{B}$ as follows:

$$\widehat{B} = \left[ G_{cls}\{P_{cls}^{F_0}, \ldots, P_{cls}^{F_T}\}; G_0\{P_0^{F_0}, \ldots, P_0^{F_T}\}; \ldots; G_N\{P_1^{F_0}, \ldots, P_N^{F_T}\} \right]. \tag{9}$$

Then, we shift and shuffle these patch groups (excluding class token group $G_{cls}$ as it will always appear in the head position) using TCSS proposed by *Alsehaim & Breckon (2022)* to make the model more robust to appearance noise (see left part of Fig. 4), which has been indicated by *Zhang et al. (2018)* and *Huang et al. (2021)*. Briefly, the first few patch groups (in the order of position, in this work the number is 2) are cut off and shifted to the end of patch groups, then, these patch groups are shuffled, the patches in latter half are inserted into the gaps between each patch in former half, as shown in the middle part of Fig. 4.

In the local part branch, as the left part of the figure shows, each feature map needs to undergo shift operation by a given amount, followed by a shuffle operation shown in the middle of the figure, then each feature map will be divided into four strips from top to bottom for separate treatment.

Subsequently, the frame bundle that has undergone shuffling is sent to part-dependent feature extraction (*Fan et al., 2020*). We divide image patches within each frame into multiple horizontal strips independently based on morphological characteristics (from top to bottom) as shown in the right part of Fig. 4. In this work, the number of strips is set to 4 due to the balance between performance and computing complexity. The features of these strips are then sent to a shared Vision Transformer Block, different from the Vision Transformer in Backbone which extracts at frame level; the block here sees the frame bundle in a unique way, like parts of aggregated frames stack. Then the shared Vision Transformer Block generates the corresponding local part features of strips, named $local_1$, $local_2$, $local_3$, and $local_4$.

### Global temporal branch

The other branch after the backbone is the global temporal branch. This branch is built specifically for the spatio-temporal feature extraction at the global level. It plays a similar role as the TP module in the common framework. Initially, at the frame level, global features $Global = [global_0; \ldots; global_T]$ are obtained using a Vision Transformer Block which works similarly to the Vision Transformer in Backbone. Then, a spatio-temporal attention which contains two convolutional layers and a final SoftMax function is applied to map the embedding dimension to 1 and generate the scores along the time dimension, *i.e.*, different gait frame (*Rao et al., 2018*), it looks like this, $Score = [score_0; \ldots; score_T]$. The final global-temporal feature $\widehat{Global}$ is generated as follows:

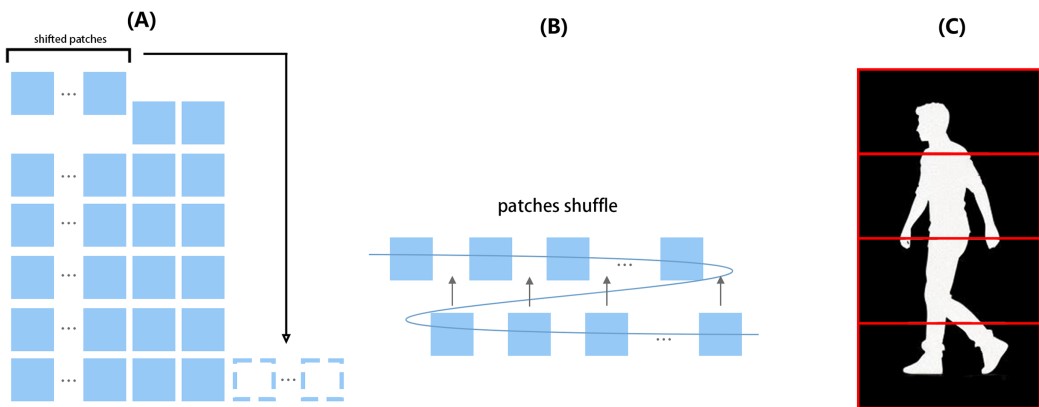

**Figure 4 Illustration of TCSS and part segmentation implemented in the local part branch.** From left to right: (A) The shifting operation extracts the first several patches from the cut images and appends them to the end of the patch sequence. (B) The patch sequence is reshaped into two rows; the second row is then inserted within the first row before reshaping back to the original form. (C) Red boxes indicate how the four strips are defined according to body parts.

$$Score = attention(Global) \tag{10}$$

$$\widehat{Global} = \sum_{i=1}^{T} score_i \odot global_i. \tag{11}$$

### BNNeck and classification head

After the extraction of global features and local part features from strips, since cross-entropy loss and triplet loss are simultaneously implemented, the model needs BNNeck proposed by *Luo et al. (2020)* to separate the features in embedding space. BNNeck adds a batch normalization layer after the generated features and before the classifier's full connection layers. They argued that many state-of-the-art methods combined ID loss and triplet loss to constrain the same feature which leads to better performance. However, better performance lets researchers ignore the inconsistency between the targets of these two losses in the embedding space. Thus, one global and four local bottlenecks, along with their linear classifiers are employed to generate $ID_{global}$, $ID_1$, $ID_2$, $ID_3$ and $ID_4$. These predicted ID labels are then sent to the optimizer along with the final features $\widehat{Global}$, $local_1$, $local_2$, $local_3$, and $local_4$.

### Loss

Inspired by *Alsehaim & Breckon (2022)*, we jointly use label smoothing cross-entropy loss $\mathcal{L}_{ce}$, triplet loss $\mathcal{L}_{triple}$, attention loss $\mathcal{L}_{att}$, and center loss $\mathcal{L}_{center}$ altogether.

$$
\begin{aligned}
L = &\mathcal{L}_{ce}\big(ID_{global}\big) + \mathcal{L}_{triple}\Big(\widehat{Global}\Big) + \beta \times \mathcal{L}_{center}\big(ID_{global}\big) \\
&+ \mathcal{L}_{att} + \frac{1}{parts}\sum_{i=1}^{parts}\big(\mathcal{L}_{ce}(ID_i) + \mathcal{L}_{triple}(local_i) + \mathcal{L}_{center}(ID_i)\big)
\end{aligned} \tag{12}
$$

where *parts* are the number of strips we split within the local part spatial branch in Fig. 2 and $\beta = 5.0 \times 10^{-5}$. Within the loss formulation Eq. (12), not only the popular gait

recognition losses already in the common framework are used, *e.g.*, label smoothing cross entropy loss and triplet loss (*Hermans, Beyer & Leibe, 2017*; *Szegedy et al., 2016*), moreover, an alternative attention loss by *Pathak, Eshratifar & Gormish (2020)* is also added for cropping out noisy frames. We also include center loss introduced by *Wen et al. (2016)* to learn more robust discriminative features with the two key objectives, inter-class dispersion and intra-class compactness as much as possible.

## GaitVViT

### Pipeline

For the proposed method GaitVViT, the model structure is shown in Fig. 5. The development of this model stems from the demand to enhance the capability of temporal information extraction within a common framework. As current research indicate, gait recognition methods emphasize temporal aspects typically by employing complex attention mechanisms or RNNs (*Dou et al., 2023*; *Zhang et al., 2019*). In the current SOTA methods, their TP modules often consist of a single layer of max pooling on the temporal dimension (*Fan et al., 2023*). These TP modules can be improved. Video ViTs treat images in sequence as frames in video, which differ from the ideas seeing them as a set and can be well-suited for this task. Thus, GaitVViT is introduced.

GaitVViT adopts the local temporal aggregation (LTA) module and global-local convolution (GLConv) module from GaitGL as the backbone (*Lin, Zhang & Yu, 2020*). So, in contrast to GaitTriViT, GaitVViT utilizes a traditional CNN as the backbone, GaitVViT takes a sequence of gait silhouette frames $Sils \in \mathbb{R}^{B \times C \times S \times H \times W}$ as inputs, where $B$ is the batch size, $C$ is the channel size, $S$ is the number of frames, and $H \times W$ are the height and width of the pre-processed gait frames. After the extraction of the CNN-based backbone, the inputs $Sils$ are mapped to a group of features $F \in \mathbb{R}^{B \times C' \times S' \times H' \times W'}$ where $C'$ is the channel size after convolution, $S'$ is length after LTA, $H' \times W'$ is the shape of each feature map. Similar to GaitTriViT, strips and part-dependent ideas are adopted, GaitVViT segments the feature maps generated by the backbone into multiple horizontal parts, shown as $F = \{P_1, P_2, \ldots, P_n\}$, where $n$ equals the number of strips. For each $P_i \in \mathbb{R}^{B \times C' \times S' \times \frac{H'}{n} \times W'}$, feature map height becomes $\dfrac{H'}{n}$ by partition. These part features are then processed by a modified Video Vision Transformer, generating the $n$ part features $F_{VViT} = \{P_1^{VViT}, P_1^{VViT}, \ldots, P_n^{VViT}\}$. For each $P_i^{VViT} \in \mathbb{R}^{B \times C' \times \frac{H'}{n} \times W'}$, the time dimension is reduced by aggregation. Subsequently, Horizontal Pooling pools each $P_i^{VViT}$ to $P_i^{HP} \in \mathbb{R}^{B \times C' \times 1}$, then model concatenates these $n$ part of $P_i^{HP}$ together at the last dimension. The final feature is shown as $P_{final}^{HP} \in \mathbb{R}^{B \times C' \times n}$. After passing through the classification head, each part will calculate its loss individually. The Eqs. (13), (14), (15) below demonstrate the brief workflow of frame $F^i$ in model GaitVViT in Fig. 5.

$$F_{conv}^{it} = Part^t(Conv(F^i)) \tag{13}$$
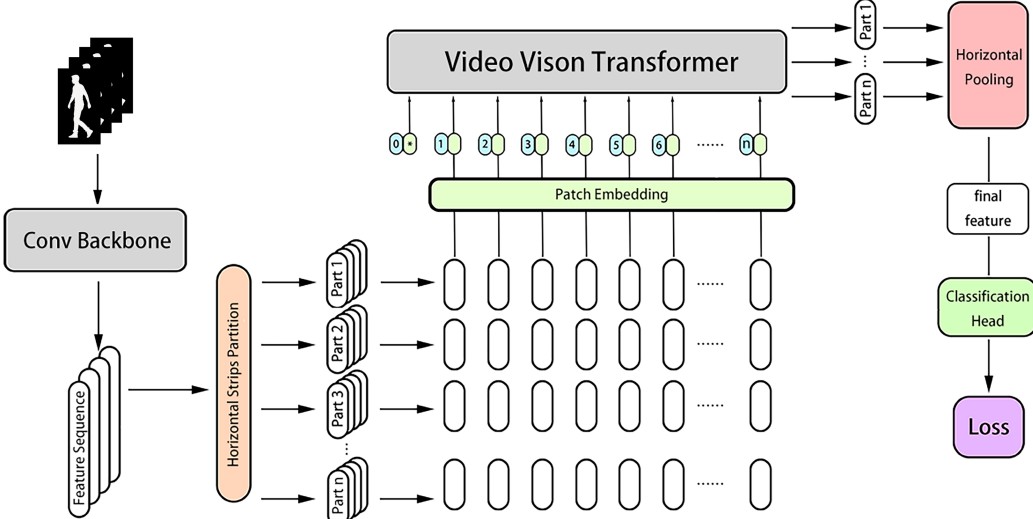

**Figure 5 The pipeline of GaitVViT including a conv backbone, a video vision transformer as temporal pooling.** Horizontal pooling, classification head and loss. From left to right: Inputs are first fed into Conv Backbone; features are cut into horizontal parts treating by Video Vision Transformer separately; the aggregated features are pooled by horizontal pooling to obtain the final features; classification head conduct batch normalization and predict the labels; losses are calculated using both triplet and cross-entropy loss.

$$F_{VViT}^i = \sum_{t=1}^{parts} VViT\left(\left[emb_{position}^{it};\ emb_{case}^{it};\ Emb\left(F_{conv}^{it}\right)\right]\right) \tag{14}$$

$$Loss_{GaitVViT}^{F^i} = L\left(Head\left(Pool\left(F_{VViT}^i\right)\right)\right). \tag{15}$$

where $Conv$ is the convolutional backbone, $Part^t$ means the t-th part of horizontal strips partition, $Emb$ is patch embedding, along with position and case embeddings, $VViT$ is the Video Vision Transformer block, $Pool$ represents horizontal pooling, $Head$ is the classification head and finally a function $L$ is used to calculate the loss.

## Backbone

In GaitVViT, a traditional CNN is implemented as the backbone, the LTA and GLConv layer proposed by *Lin et al. (2022)* are also adopted. The overview of the backbone structure is shown in Fig. 6. The CNN-based backbone consists of multiple convolutional layers. At first, each input will be extracted by a 3DCNN layer with a kernel size of $[3 \times 3 \times 3]$ to obtain shallow features. Next, the LTA operation is employed to aggregate the temporal information and preserve more spatial information for trade-off. After that, Global and local feature extractor layers are implemented which consist of the GLConvA0 layer, Max Pooling layer, GLConvA1 layer, and GLConvB0 layer. The max pooling operation is implemented to down-sample the feature size at the last two dimensions for computing complexity trade-off. After the extractor, the combined feature assembling both global and local information is generated.

**Figure 6 The structure of convolutional backbone in GaitVViT with multiple convolutional layers.**
From left to right: 3DCNN layer, local temporal aggregation (LTA), Global and local extractor consists of
GLConvA0, max pooling layer, GLConvA1 and GLConvB0.

The details of the global and local convolutional layer (GLConv) are shown in Fig. 7. It
basically consists of two parallel paths: one for local feature extraction and one for global
feature extraction, which can take advantage of both global and local information.

The feature map will go through two branches. The upper branch is for local extraction
where feature maps need partition before 3D convolution, the branch below is global
extraction takes the whole feature map as input. And there are two combination methods:
element-wise addition and concatenation.

The global branch implements a basic 3DCNN layer. It extracts the whole gait
information and pays attention to the relations among local regions. The local branch is
basically a 3D version of the Focal Convolutional layer proposed by *Fan et al. (2020)*. It
implements a 3DCNN layer with a shared kernel, the feature map will be split into several
parts before the 3DCNN layer. They extract the local features and then combine them,
which contain more detailed information than the global gait features. GLConv has two
different structures due to different combinations between global and local features,
GLConvA uses element-wise addition and GLConvB uses concatenation.

### Video vision transformer encoder

After the extraction of the backbone. Feature maps exist in the form of
$F = \{P_1, P_2, \ldots, P_n\}$, where $n$ equals the number of strips. For each $P_i \in \mathbb{R}^{B \times C' \times S' \times \frac{H'}{n} \times W'}$
GaitVViT conducts the temporal aggregation individually.

Original Vision Transformer (ViT) regards an image as a grid of non-overlapping
patches (*Dosovitskiy et al., 2020*), thus, the transformer extracts the features of each patch
and constrains the spatial connection inter-patch. The modified Video Vision Transformer
(VViT) regards each frame in a sequence as an independent patch, and the multi-head
self-attention among spatial patches in the original ViT can be smoothly transferred to a
temporal attention learning the connection among each frame. Researchers have
implemented VViT-based methods in many video-based recognition tasks, *e.g.*, ViViT by
*Arnab et al. (2021)*, Video Transformer Network by *Neimark et al. (2021)* and Video Swin
Transformer by *Liu et al. (2021)*.

GaitVViT adopted a modified LongFormer as the specific Video Vision Transformer
Encoder (*Beltagy, Peters & Cohan, 2020*). The LongFormer in the encoder leverages sliding
window to preserve the edge information between adjacent frames and strengthen the
inter-frame connections. Before the Video Vision Transformer Encoder, part feature $P_i$
will rearrange to $P_i^{pre} \in \mathbb{R}^{B \times S' \times \left(\frac{H'}{n} \times W' \times C'\right)}$, then, all part features will be concatenated at the

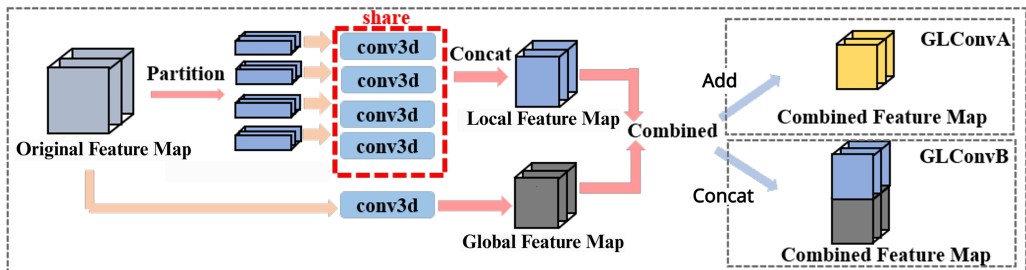

**Figure 7 The structure of GLConv layer as a two branches path for the feature map.** An example module in the backbone, where 'conv3d' denotes 3D CNN layers. During local feature extraction, the 'conv3d' layers for different strips share weights. Global and local features can be combined through addition or concatenation.

first dimension to form the $F_{pre} \in \mathbb{R}^{(B \times n) \times S' \times \left(\frac{H'}{n} \times W' \times C'\right)}$. After the temporal extraction of encoder, the second dimension of $F_{pre}$ is reduced, the aggregated feature $F_{post} \in \mathbb{R}^{(B \times n) \times \left(\frac{H'}{n} \times W' \times C'\right)}$ will be rearranged back to $F_{VViT} = \left\{P_1^{VViT}, P_2^{VViT}, \ldots, P_n^{VViT}\right\}$, where $P_i^{VViT} \in \mathbb{R}^{B \times C' \times \frac{H'}{n} \times W'}$.

### Classification head and loss

Like GaitTriViT, the final feature will be fed into a Batch Normalization layer followed by a fully connected layer to generate the predicted labels. Both triplet loss and cross-entropy loss are employed to optimize the model. The triplet losses are calculated between feature anchors, and the cross-entropy losses are calculated on the predicted label matrix.

## Summary

In this section, two Transformer-based architectures are proposed for gait recognition, GaitTriViT and GaitVViT. For GaitTriViT, the Vision Transformer is used as the frame-level backbone while incorporating case embedding and angle embedding to enhance frame-level feature extraction performance. Taking into account the similarity between gait recognition tasks and person re-identification tasks, this work draws inspiration from several articles on ReID tasks and introduces TCSS by *Alsehaim & Breckon (2022)*, as well as the combination of part-dependent strategy (*Fan et al., 2020*), dividing frame-level features into different strips before another Vision Transformer block. These components above collectively build the local part spatial branch after the backbone, dedicated to extracting local spatial features. Another branch after the backbone employs another Vision Transformer block to extract global features, where temporal attention is used to jointly learn global temporal features (*Fu et al., 2019*; *Rao et al., 2018*; *Zhang et al., 2020*). The features from both branches are combined to generate the final features, and the predicted labels are generated by classification heads. For the optimizer, multiple loss functions are introduced to optimize the model together.

For GaitVViT, given the gait recognition common framework (*Fan et al., 2023*), where the wildly implemented TP module often consists of a single max pooling layer, which will waste the sequence information and need more attention for a complete improvement. The Video ViT is a variant of the original Vision Transformer (*Dosovitskiy et al., 2020*). Video ViT is created from the idea that regarding every frame in video as a patch. In the traditional transformer structure, every patch is a non-overlapping square region of an image, so when we change the scale and arrangement, the transformer can conduct the extraction on a whole sequence and run the self-attention on the time dimension. Adopting the LTA and GLConv layers from GaitGL proposed by *Lin et al. (2022)*, this work connects the extracted feature representations to a Video ViT Encoder. The encoder implemented by LongFormer will conduct temporal extraction and aggregate the inputs to obtain the final features (*Beltagy, Peters & Cohan, 2020*).

## IMPLEMENTATION

### Introduction

In this section, the datasets and implementation details are discussed, including two popular benchmarks, CASIA-B and OUMVLP (*Takemura et al., 2018*; *Yu, Tan & Tan, 2006*). Moreover, several details during the training and evaluation phase are explained.

### Datasets

**CASIA-B** is provided by *Yu, Tan & Tan (2006)* for gait recognition and to promote research. CASIA-B is a large multi-view gait database, which was created in January 2005. It has 124 subjects, and the gait data was captured from 11 views. Three variations, namely view angle, clothing and carrying condition changes, are separately considered. In this article, we use the human silhouettes extracted from video files as benchmarks. The format of the filenames in CASIA-B is 'xxx-mm-nn-ttt.png', where 'xxx' is subject id, 'mm' stands for walking status, including 'nm' (normal), 'cl' (in a coat) or 'bg' (with a bag), 'nn' is sequence number for each walking status, normal walking has six sequences, wearing coat and carrying bag have two sequences each; 'ttt' is view angle can be '000', '018', …, '180' . Examples of CASIA-B are shown in Fig. 8.

Each subject has a maximum of 110 sequences. We use subjects with ID from 1 to 74 as the training set, and subjects with ID from 75 to 124 as the test set. During the testing phase, we use the first four sequences from 'nm' (nm-1, nm-2, nm-3, nm-4) as the gallery set, and the remaining six sequences are divided into three query sets based on their respective situations: 'nm' query includes 'nm-5' and 'nm-6', 'bg' query includes 'bg-1' and 'bg-2', and 'cl' query includes 'cl- 1' and 'cl-2' (*Chao et al., 2018*; *Fan et al., 2020*; *Lin et al., 2022*; *Yu, Tan & Tan, 2006*).

**OUMVLP** is part of the OU-ISIR Gait Database, which stands for Multi-View Large Population Dataset, provided by *Takemura et al. (2018)*. OUMVLP is meant to aid research efforts in the general area of developing, testing and evaluating algorithms for cross-view gait recognition. The Institute of Scientific and Industrial Research (ISIR), Osaka University (OU) has copyright in the collection of gait video and associated data

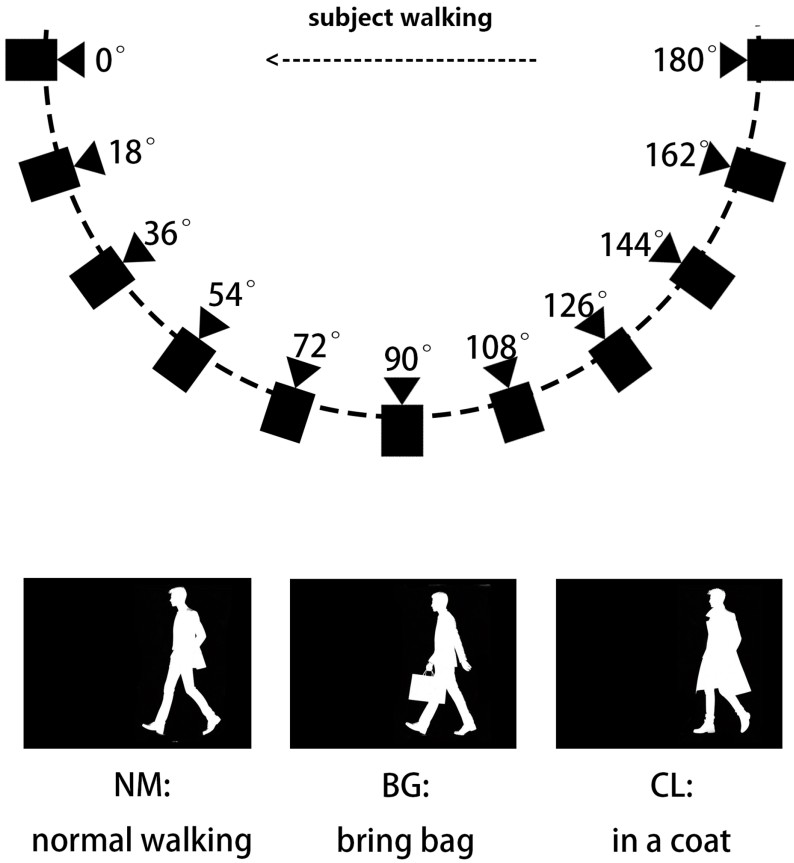

**Figure 8 The illustration of silhouettes from different camera angles and walking status in the CASIA-B dataset.** The CASIA-B dataset has 124 subjects. The silhouettes are shot from 11 camera angles and three different walking status. 'BG', 'CL' and 'NM' stand for three different walking status: Bring Bag, In a Coat and Normal Walking.

and serves as a distributor of the OU-ISIR Gait Database. The data was collected in conjunction with an experience-based long-run exhibition of video-based gait analysis at a science museum. The dataset consists of 10,307 subjects (5,114 males and 5,193 females with various ages, ranging from 2 to 87 years) from 14 view angles, ranging 0°–90°, 180°–270°. Gait images of 1,280 × 980 pixels at 25 fps are captured by seven network cameras (Cam1–7) placed at intervals of 15-degree azimuth angles along a quarter of a circle whose center coincides with the center of the walking course. The illustration is shown in Fig. 9. Each subject has two sequences, 00 for probe and 01 for gallery. We select 5,153 subjects with odd-numbered IDs as the training set, and the remaining 5,154 subjects as the test set (*Chao et al., 2018*; *Fan et al., 2020*; *Lin et al., 2022*; *Takemura et al., 2018*).

## Implementation details

The project is implemented on both Windows 11 and Debian, but mainly on Windows 11. The Python IDE chosen is PyCharm 2021.1.3.0 and Python version is 3.9.0. The framework chosen is PyTorch 1.13.0. The project is trained and evaluated on one NVIDIA GeForce RTX 3080 Ti GPU with CUDA version 11.8.

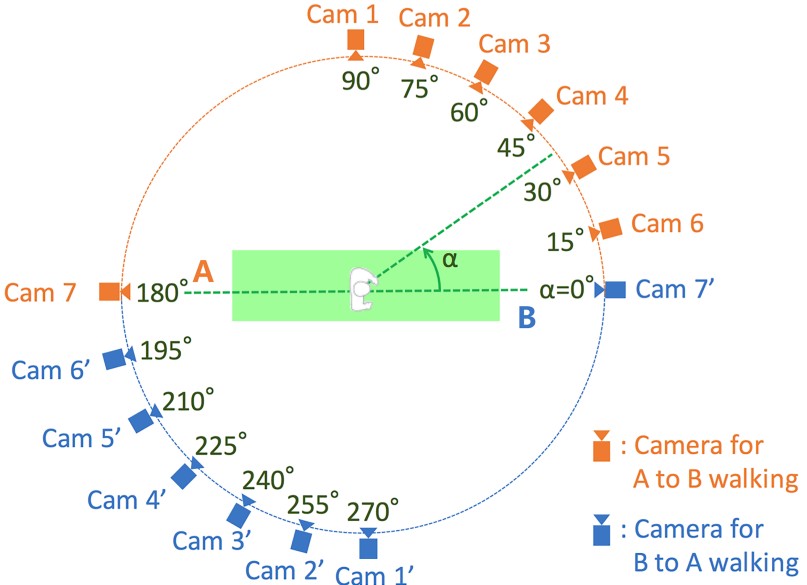

**Figure 9** **Illustration of silhouettes from different camera angles and how they are shot in OUMVLP dataset.** OUMVLP has 10,307 subjects. Silhouettes are shot from 14 different angles, which don't have different walking status between sequences.

We follow the common way by *Fan et al. (2022)* to pre-process the silhouette data from CASIA-B and OUMVLP datasets, then choose the hyperparameters taking hardware limitations and efficiency into consideration. This pre-processing involved removing invalid data not having whole body in frame, arranging frames in structure of ID-condition-angle, aligning and cropping silhouette images to ensure the subject's body is in the center of the image, and has a proper size, more details can refer to original works by *Fan et al. (2022)*. After pre-processing, each frame image's size is $64 \times 44$. For GaitTriViT specifically, since we don't have the computational resource to train a well performed image model from scratch, we initialized the Vision Transformer backbone with parameters pre-trained on ImageNet-21K (*Deng et al., 2009*; *Wightman, 2019*), the input of ViT requires RGB-like images with three channels and a size of $256 \times 128$. But our silhouettes are single-channel binary images. Therefore, we inserted a fully connected layer in the head of backbone with an input dimension of 1 and an output dimension of 3 to map the silhouette from $Sil \in \mathbb{R}^{H \times W \times 1}$ to $\widehat{Sil} \in \mathbb{R}^{H \times W \times 3}$ pseudo-RGB images. And in data augmentation phase, we resized the images to the required size.

CASIA-B and OUMVLP datasets differ in camera angles and walking scenarios. CASIA-B has 11 camera angles and a total of 10 walking sequences. Therefore, in Eq. (3), $E_{angle}$ has the shape of $[a \times D]$, where $a = 11$, and $E_{case}$ is in shape of $[c \times D]$, where $c = 10$. $D$ is the embedding dimension set to 768. In contrast, OUMVLP has 14 camera angles and no distinction in walking scenarios, so only $E'_{angle} \in \mathbb{R}^{a' \times D}$, where $a' = 14$.

In this work, excluding the ViT in backbone of GaitTriViT, most parameters are initialized using the Kaiming initialization (*He et al., 2015*) following the VID-Trans-ReID method. For GaitTriViT, the number of frames T in a frame bundle is set to 4. The

selection strategy of frames in every bundle when training is dividing the whole sequence into T parts and randomly selecting one frame from each part, creating a frame bundle where each frame can be selected again. During testing, T frames are sequentially selected from the whole sequence. The batch size is set to 52, the optimizer is stochastic gradient descent (SGD), and the scheduler is using cosine learning rate decay with warming up (*Loshchilov & Hutter, 2017*). For GaitVViT, during the training phase, the number of frames T in each batch is set to 30, the selection strategy is randomly choosing 30 frames in order among the sequence. During the test phase, the model uses all frames within one sequence to generate the final feature. The batch size is set to 36, the optimizer is Adam, and the scheduler is multi step learning rate.

OUMVLP has 10,307 subjects, silhouettes are shot from 14 different angles, which don't have different walking status between sequences.

## Summary

In this section, the choices of datasets and implementation details are explained. Two popular datasets are chosen in this work: CASIA-B and OUMVLP. CASIA-B is a classic dataset for gait recognition research specifically. It has 124 subjects; each subject has multiple sequences varied in 11 camera angles and three walking statuses. OUMVLP is a new dataset compared to CASIA-B, it has the most subjects among the gait datasets so far, which consists of 5,114 males and 5,193 females captured from 14 camera angles, each subject has two sequences. Furthermore, several implementation details are explained including pre-processing, different settings on each benchmark and details in hyper-parameters.

## EVALUATION

### State-of-the-art comparison

In gait recognition tasks, many researchers have made a lot of contributions. The GEINet by *Shiraga et al. (2016)* leverages the gait energy images (GEI) as the representations of gait features, open the research towards gait recognition. The GaitSet by *Chao et al. (2018)* led the new era of appearance-based Gait Recognition, then plenty of novel models came out *e.g.*, GaitPart by *Fan et al. (2020)*, GaitGL by *Lin, Zhang & Yu (2020)*, GaitBase by *Fan et al. (2022)*, SRN by *Hou et al. (2021)*, GLN by *Hou et al. (2020)* and DeepGait-3D by *Fan et al. (2023)*. They are all regarded as state-of-the-art models by now, in which GaitSet, GaitPart and GaitGL are considered most iconic and are chosen mostly as milestones.

The evaluation metric of single-view-gallery-evaluation is used in almost all silhouette-based gait recognition methods. First, the test set needs to split all the sequences into two subsets named probe and gallery, where each subject in probe has a sequence consist of M camera views and N for gallery, where M is equal to N in most cases. Each probe-gallery view pair is denoted as $\left(v_i^p, v_j^g\right)$, where $i \in [1, 2, \ldots, M], j \in [1, 2, \ldots, N]$.

Second, for each probe-gallery view pair $\left(v_i^p, v_j^g\right)$, after model inference $\mathcal{J}$, every probe sequence under the view $V_i^p$ is compared to all gallery sequences under different view $V_j^g$ to calculate the Euclidean distance, the label of the closest gallery sequence will be assigned to the probe. The rank-1 accuracy $ACC_{ij}$ is then computed for each probe-gallery pair by

checking the percentage of the predicted label matching the ground-truth *label*. In the end, an accuracy matrix *ACC* in the shape of $M \times N$ is obtained.

Third, the identical-view cases, where $V_i^p = V_j^g$, are always excluded. So, in the *ACC* matrix, the corresponding elements (mostly the diagonal elements) are abandoned while the rest are averaged as the evaluation metric. At last, the accuracy for each probe view $ACC_i^p$ is acquired by averaging the elements of *ACC* in row excluding identical-view cases. The processes are as follows:

$$ACC_{ij} = R1\left(euc\left(\mathcal{J}\left(V_i^p\right), \ \mathcal{J}\left(V_j^g\right)\right), \ label\right) \tag{16}$$

$$ACC_i^p = \frac{1}{N-1}\left(\sum_{j=1}^{N} ACC_{ij} - ACC_{ii}\right) \tag{17}$$

where *R1* stands for rank-1 calculation and *euc* is short of Euclidean Distance.

In this article, the 'Single' and 'Cross' marks in the following tables indicate the different evaluation protocols. The 'Single' stands for the single-view-gallery evaluation which is the regular evaluation method explained above. For example, the CASIA-B dataset has 3 walking status and 11 camera angles (*Yu, Tan & Tan, 2006*), so, for probe sequences whose walking status is 'NM' and view angle is '090', they needed to compare with 10 galleries with different view angle excluding the gallery having the same view, the average of 10 results become the final result of this specific probe. The 'Cross' stands for cross-view-gallery evaluation. Particularly, for each probe view, the sequences of all gallery views are adopted for the comparison with the identical-view cases excluded. The accuracy under cross-view-gallery evaluation is quite higher than single-view-gallery, because subjects in some views may experience significant silhouette changes, bringing difficulty and less discriminativeness for recognition (*Hou et al., 2023*).

The evaluation results of two proposed methods GaitTriViT and GaitVViT are presented below, as shown in Tables 1 and 2. The data of the state-of-the-art methods are collected from their own articles.

The experiments show that GaitTriViT faces huge difficulties with the two popular benchmarks. The regular single-view-gallery accuracy can only surpass the GEINet, indicating the bad generalization of GaitTriViT. Even the cross-view-gallery performances are dropped when the walking status is bringing a bag, or especially, wearing a coat. The GaitTriViT has bad robustness towards appearance noises.

GaitVViT performs better, on CASIA-B, when the walking status is normal walking, the performance of GaitVViT can slightly surpass GaitGL at probe views of 0°, 18°, 36°, 54° and 126°, making the average accuracy slightly better too. But it doesn't perform well enough for a proposed transformer-enhanced method when the walking status is bag bring or wearing a coat. Maybe due to the sensitivity of transformer-based structure for appearance information, *i.e.*, the method is less robust to appearance noises.

Comparison shows that GaitTriViT focus more on spatial feature extraction by employing three individual vision transformer in each extraction phase (one for frame-level features in the backbone, one for set-level local features in a local branch, and

**Table 1  State-of-the-art comparison on OUMVLP.** Rank-1 accuracy in 14 probe view angle, excluding identical-view cases.

| Evaluation | Method | Probe view | | | | | | | | | | | | | | Mean |
|---|---|---|---|---|---|---|---|---|---|---|---|---|---|---|---|---|
| | | 0° | 15° | 30° | 45° | 60° | 75° | 90° | 180° | 195° | 210° | 225° | 240° | 255° | 270° | |
| Single | GaitSet | 79.30 | 87.90 | 90.00 | 90.10 | 88.00 | 88.70 | 87.70 | 81.80 | 86.50 | 89.00 | 89.20 | 87.20 | 87.60 | 86.20 | 87.10 |
| | GaitPart | 82.60 | 88.90 | 90.80 | 91.00 | 89.70 | 89.90 | 89.50 | 85.20 | 88.10 | 90.00 | 90.10 | 89.00 | 89.10 | 88.20 | 88.70 |
| | GLN | 83.81 | 90.00 | 91.02 | 91.21 | 90.25 | 89.99 | 89.43 | 85.28 | 89.09 | 90.47 | 90.59 | 89.60 | 89.31 | 88.47 | 89.18 |
| | GaitGL | 84.90 | 90.20 | 91.10 | 91.50 | 91.10 | 90.80 | 90.30 | 88.50 | 88.60 | 90.30 | 90.40 | 89.60 | 89.50 | 88.80 | 89.70 |
| | GaitBase | – | – | – | – | – | – | – | – | – | – | – | – | – | – | 90.80 |
| | DeepGait-3D | – | – | – | – | – | – | – | – | – | – | – | – | – | – | 92.00 |
| | GaitTriViT | 58.32 | 72.90 | 80.30 | 82.15 | 76.27 | 75.85 | 73.36 | 59.53 | 73.45 | 79.62 | 81.32 | 75.70 | 75.37 | 72.16 | 74.02 |
| | GaitVViT | 81.20 | 88.95 | 90.26 | 90.54 | 89.02 | 89.27 | 88.40 | 85.34 | 87.42 | 88.98 | 89.26 | 87.54 | 87.86 | 86.51 | 87.90 |
| Cross | GaitTriViT | 84.52 | 97.71 | 98.73 | 98.67 | 98.43 | 99.50 | 99.52 | 88.84 | 98.09 | 98.68 | 98.83 | 98.65 | 99.32 | 99.45 | 97.07 |

**Note:**
'Single' and 'Cross' are two different evaluation protocols. GaitVViT reaches the level of GaitSet, whereas GaitTriViT experiences significant performance decline when probe view is 0° and 180° in Single View protocol.

one for frame-level extraction in the global branch before the attention module), while the specific temporal modeling task is assigned to a spatio-temporal attention module; in contrast, GaitVViT adopted the Video ViT to replace the common TP module and enhance its functionality, which focus more in temporal aspect apparently. Given the situation that current Transformer-based methods have not achieved astonishing outcomes in the field of gait recognition, the current ViT structure may not be a good upstream backbone for inputs like gait silhouette. According to the argument by *Fan et al. (2023)*, many patches on a gait silhouette are all-white (all 1) or all-black (all 0), where neither posture nor appearance information are provided. They call them dumb patches. Since all values from a dumb patch are all 0 or all 1, These all-1 or all-0 dumb patches can make backward gradients significantly ineffective or even computationally invalid for the parameter optimization of downstream ViT layers. For GaitVViT, it meets the basic line of current state-of-the-art methods. A traditional CNN backbone makes sure the performance away from too bad, despite the augmentation on temporal pooling module gains no astonishing improvement. Also, these two methods lack in-wild testing. Gait recognition methods on pre-processed datasets, to some extent, have already saturated their performance, application performance in the real world will be more important in future research (*Cosma, Catruna & Radoi, 2023*; *Cosma & Radoi, 2021*).

### Ablation study

In this work, multiple technologies are employed in two methods. For GaitTriViT, there are TCSS and angle embedding (also case embedding on the CASIA-B dataset). But it achieves not a promising performance on two popular benchmarks. Several ablation experiments were carried to deep dive into the contribution of each technology and try to improve performance by introducing extra mechanisms, *e.g.*, excluding specific modules, changing the selection method, rearranging the order of strip segmentation, and introducing part embeddings. For GaitVViT, there is also an ablation study excluding

**Table 2 Rank-1 accuracy comparison on CASIA-B in three walking status and 11 view angle, excluding identical-view cases.**

| Evaluation | Status | Model | Probe view | | | | | | | | | | | Mean |
|---|---|---|---|---|---|---|---|---|---|---|---|---|---|---|
| | | | 0° | 18° | 36° | 54° | 72° | 90° | 108° | 126° | 144° | 162° | 180° | |
| Single | NM | GEINet | 56.10 | 69.10 | 76.20 | 74.80 | 68.50 | 65.60 | 70.80 | 78.00 | 75.60 | 68.40 | 57.50 | 69.15 |
| | | GaitSet | 90.80 | 97.90 | 99.40 | 96.90 | 93.60 | 91.70 | 95.00 | 97.80 | 98.90 | 96.80 | 85.80 | 95.00 |
| | | GaitPart | 94.10 | 98.60 | 99.30 | 98.50 | 94.00 | 92.30 | 95.90 | 98.40 | 99.20 | 97.80 | 90.40 | 96.20 |
| | | GLN | 93.20 | 99.30 | 99.50 | 98.70 | 96.10 | 95.60 | 97.20 | 98.10 | 99.30 | 98.60 | 90.10 | 96.88 |
| | | GaitGL | 96.00 | 98.30 | 99.00 | 97.90 | 96.90 | 95.40 | 97.00 | 98.90 | 99.30 | 98.80 | 94.00 | 97.40 |
| | | GaitBase | | | | | | | | | | | | 97.60 |
| | | GaitTriViT | 78.40 | 84.20 | 91.10 | 86.70 | 78.30 | 77.80 | 81.50 | 87.00 | 91.00 | 85.50 | 76.50 | 83.45 |
| | | GaitVViT | 96.50 | 99.20 | 99.40 | 98.20 | 96.90 | 94.00 | 96.70 | 99.40 | 99.30 | 98.30 | 93.60 | 97.41 |
| | BG | GEINet | 44.80 | 53.64 | 54.55 | 51.73 | 49.40 | 46.60 | 47.30 | 56.50 | 58.20 | 49.90 | 45.10 | 50.70 |
| | | GaitSet | 83.80 | 91.20 | 91.80 | 88.80 | 83.30 | 81.00 | 84.10 | 90.00 | 92.20 | 94.40 | 79.00 | 87.20 |
| | | GaitPart | 89.10 | 94.80 | 96.70 | 95.10 | 88.30 | 84.90 | 89.00 | 93.50 | 96.10 | 93.80 | 85.80 | 91.50 |
| | | GLN | 91.10 | 97.68 | 97.78 | 95.20 | 92.50 | 91.20 | 92.40 | 96.00 | 97.50 | 94.95 | 88.10 | 94.04 |
| | | GaitGL | 92.60 | 96.60 | 96.80 | 95.50 | 93.50 | 89.30 | 92.20 | 96.50 | 98.20 | 96.90 | 91.50 | 94.50 |
| | | GaitBase | | | | | | | | | | | | 94.00 |
| | | GaitTriViT | 71.00 | 74.50 | 78.80 | 76.26 | 67.70 | 65.20 | 68.80 | 77.20 | 79.90 | 77.07 | 66.70 | 73.01 |
| | | GaitVViT | 90.50 | 95.60 | 95.90 | 93.64 | 89.30 | 82.40 | 88.20 | 94.30 | 96.30 | 94.04 | 90.80 | 91.91 |
| | CL | GEINet | 21.80 | 30.90 | 36.30 | 34.40 | 35.90 | 30.20 | 31.10 | 32.10 | 28.90 | 23.80 | 25.90 | 30.12 |
| | | GaitSet | 61.40 | 75.40 | 80.70 | 77.30 | 72.10 | 70.10 | 71.50 | 73.50 | 73.50 | 68.40 | 50.00 | 70.40 |
| | | GaitPart | 70.70 | 85.50 | 86.90 | 83.30 | 77.10 | 72.50 | 76.90 | 82.20 | 83.80 | 80.20 | 66.50 | 78.70 |
| | | GLN | 70.60 | 82.40 | 85.20 | 82.70 | 79.20 | 76.40 | 76.20 | 78.90 | 77.90 | 78.70 | 64.30 | 77.50 |
| | | GaitGL | 76.60 | 90.00 | 90.30 | 87.10 | 84.50 | 79.00 | 84.10 | 87.00 | 87.30 | 84.40 | 69.50 | 83.60 |
| | | GaitBase | | | | | | | | | | | | 77.40 |
| | | GaitTriViT | 27.10 | 32.20 | 40.50 | 46.50 | 45.60 | 40.90 | 42.20 | 42.50 | 40.60 | 28.60 | 25.50 | 37.47 |
| | | GaitVViT | 67.20 | 81.70 | 86.20 | 82.30 | 76.90 | 70.50 | 75.30 | 80.50 | 84.30 | 80.20 | 62.50 | 77.05 |
| Cross | NM | GEINet | 92.00 | 94.00 | 96.00 | 90.00 | 100.0 | 98.00 | 100.0 | 93.88 | 89.80 | 83.67 | 81.63 | 92.63 |
| | | GaitSet | 100.0 | 100.0 | 100.0 | 100.0 | 100.0 | 100.0 | 100.0 | 100.0 | 100.0 | 100.0 | 100.0 | 100.0 |
| | | SRN | 100.0 | 100.0 | 100.0 | 100.0 | 100.0 | 100.0 | 100.0 | 100.0 | 100.0 | 100.0 | 100.0 | 100.0 |
| | | GaitTriViT | 100.0 | 100.0 | 100.0 | 100.0 | 100.0 | 100.0 | 100.0 | 100.0 | 100.0 | 100.0 | 99.00 | 99.91 |
| | BG | GEINet | 80.00 | 94.00 | 90.00 | 87.76 | 94.00 | 94.00 | 92.00 | 94.00 | 90.00 | 84.00 | 82.00 | 89.25 |
| | | GaitSet | 100.0 | 98.00 | 98.00 | 97.96 | 98.00 | 98.00 | 98.00 | 100.0 | 100.0 | 100.0 | 100.0 | 98.91 |
| | | SRN | 100.0 | 100.0 | 100.0 | 100.0 | 100.0 | 100.0 | 100.0 | 100.0 | 100.0 | 100.0 | 100.0 | 100.0 |
| | | GaitTriViT | 95.00 | 89.00 | 94.00 | 93.94 | 97.00 | 93.00 | 93.00 | 97.00 | 94.00 | 89.90 | 86.00 | 92.89 |
| | CL | GEINet | 84.00 | 94.00 | 94.00 | 88.00 | 94.00 | 96.00 | 98.00 | 100.0 | 92.00 | 88.00 | 86.00 | 92.18 |
| | | GaitSet | 98.00 | 100.0 | 100.0 | 100.0 | 100.0 | 100.0 | 100.0 | 100.0 | 100.0 | 100.0 | 100.0 | 99.82 |
| | | SRN | 100.0 | 100.0 | 100.0 | 100.0 | 100.0 | 100.0 | 100.0 | 100.0 | 100.0 | 100.0 | 100.0 | 100.0 |
| | | GaitTriViT | 37.00 | 41.00 | 49.00 | 60.00 | 69.00 | 67.00 | 60.00 | 59.00 | 49.00 | 30.00 | 32.00 | 50.27 |

**Note:**
'Single' and 'Cross' are two evaluation protocols. GaitVViT reaches the same level of SOTA, while GaitTriViT experiences significant performance decline when subjects walking status is 'CL', either in 'Single' or 'Cross' protocol.

certain modules or modifications, to explore the individual contribution of each mechanism and potential of the model.

Some other implementation details may also have an influence on model performance. While this work using Kaiming initialization and ImageNet pretraining, as well as a combination of loss functions (*e.g.*, triplet, cross-entropy, and attention loss), we completely follow the steps of original articles, where more details will be demonstrated. Exploring alternative strategies may potentially contribute to the performance of gait recognition. For example, *Palla, Parida & Sahu (2024)* introduced a hybrid Harris hawks and Arithmetic optimization algorithm (*Heidari et al., 2019*; *Rao et al., 2023*) introduced a hybrid whale and gray wolf optimization algorithm (*Mirjalili & Lewis, 2016*; *Mirjalili, Mirjalili & Lewis, 2014*), trying to select the most optimal gait features. They argue the optimization problems have become increasingly complex, traditional mathematical methods (*e.g.*, Newton's descent and gradient descent method) become difficult solving them effectively. Thus, many scholars focus on meta heuristics (MAs), a class of optimization algorithms inspired by natural phenomena (*e.g.*, biological group behavior, physical phenomena and evolutionary laws). MAs require no gradient information, flexible, easily implemented, and widely used in complex situations (*Jia et al., 2023*). These MAs can also improve the performance compared to the mathematic loss optimization used now.

If not mentioned, the results below are obtained following the cross-view-gallery evaluation, as it is closer to real-world application scenarios.

### Analysis of excluding specific module

For GaitTriViT, given the utilization of multiple techniques in the proposed method and the observed insufficient model performance, understanding the individual contributions or potential hindrances of each technology becomes essential. Thus, there are pairs of tests with different situations on OUMVLP and CASIA-B, *e.g.*, no TCSS, no angle embedding or neither (*Alsehaim & Breckon, 2022*). The evaluation results are shown in Tables 3 and 4.

The test results in Table 3 show that removing the Temporal Clips Shift and Shuffle (TCSS) module during inference could slightly improve the performance when the camera angle is not near 90° and 270°. Maybe because the TCSS module shuffles the images, which introduces extra noises to appearance information, making the gait sequences less discriminative. But gait sequences near 90° and 270° show the side of subjects on silhouette, which usually contain more gait patten information than appearance information in ratio, so the model will be more robust towards appearance noises and put more attention on generating distinct gait features. Also in Table 3, when angle embeddings are removed, the more probe view close to 0° and 180°, the more significantly the test accuracy is dropped, while the accuracy drop from probe view near 90° and 270° can be almost ignored. The results also indicate the side silhouettes contain more discriminative information for better inference performance, while the probe sequences away from side angle need angle embeddings to augment the feature representations.

For the test on CASIA-B, as shown in Table 4, the cross-view-gallery accuracy observes no changes when the modules are removed individually. Only when both modules are

**Table 3 GaitTriViT's Rank-1 accuracy on OUMVLP in 14 probe view angle, excluding identical-view cases.**

| Method | Probe view | | | | | | | | | | | | | | Mean |
|---|---|---|---|---|---|---|---|---|---|---|---|---|---|---|---|
| | 0° | 15° | 30° | 45° | 60° | 75° | 90° | 180° | 195° | 210° | 225° | 240° | 255° | 270° | |
| No TCSS | 84.29 | 97.65 | 98.73 | 98.71 | 98.43 | 99.48 | 99.52 | 88.62 | 98.07 | 98.66 | 98.83 | 98.62 | 99.30 | 99.45 | 97.03 |
| No emb | 41.87 | 79.27 | 89.87 | 86.25 | 88.24 | 98.29 | 98.89 | 49.63 | 77.84 | 89.77 | 87.91 | 93.29 | 98.16 | 98.88 | 84.15 |
| Baseline | 82.68 | 97.67 | 98.63 | 98.63 | 98.63 | 99.54 | 99.52 | 87.85 | 97.69 | 98.46 | 98.79 | 98.54 | 99.34 | 99.47 | 96.82 |

**Note:**
Results show in datasets with single walking status, removing TCSS can slightly improve the performance.

**Table 4 GaitTriViT's Rank-1 accuracy on CASIA-B in 11 probe view angle and 3 walking status, excluding identical-view cases.**

| Status | Method | Probe view | | | | | | | | | | | Mean |
|---|---|---|---|---|---|---|---|---|---|---|---|---|---|
| | | 0° | 18° | 36° | 54° | 72° | 90° | 108° | 126° | 144° | 162° | 180° | |
| NM | no TCSS | 100.0 | 100.0 | 100.0 | 100.0 | 100.0 | 100.0 | 100.0 | 100.0 | 100.0 | 100.0 | 98.00 | 99.82 |
| | no emb | 100.0 | 100.0 | 100.0 | 100.0 | 100.0 | 100.0 | 100.0 | 100.0 | 100.0 | 100.0 | 98.00 | 99.82 |
| | both neither | 100.0 | 100.0 | 100.0 | 99.00 | 100.0 | 100.0 | 100.0 | 100.0 | 100.0 | 97.00 | 95.00 | 99.18 |
| | baseline | 100.0 | 100.0 | 100.0 | 100.0 | 100.0 | 100.0 | 100.0 | 100.0 | 100.0 | 100.0 | 99.00 | 99.91 |
| BG | no TCSS | 95.00 | 89.00 | 94.00 | 93.94 | 96.00 | 93.00 | 93.00 | 97.00 | 94.00 | 89.90 | 86.00 | 92.80 |
| | no emb | 95.00 | 89.00 | 95.00 | 92.93 | 96.00 | 93.00 | 93.00 | 97.00 | 94.00 | 90.91 | 85.00 | 92.80 |
| | both neither | 88.00 | 87.00 | 89.00 | 88.89 | 94.00 | 93.00 | 93.00 | 94.00 | 90.00 | 82.83 | 81.00 | 89.16 |
| | baseline | 95.00 | 89.00 | 94.00 | 93.94 | 97.00 | 93.00 | 93.00 | 97.00 | 94.00 | 89.90 | 86.00 | 92.89 |
| CL | no TCSS | 37.00 | 35.00 | 44.00 | 61.00 | 68.00 | 63.00 | 60.00 | 57.00 | 50.00 | 32.00 | 30.00 | 48.82 |
| | no emb | 37.00 | 35.00 | 44.00 | 62.00 | 68.00 | 64.00 | 60.00 | 57.00 | 50.00 | 31.00 | 30.00 | 48.91 |
| | both neither | 21.00 | 21.00 | 23.00 | 28.00 | 46.00 | 43.00 | 43.00 | 32.00 | 21.00 | 20.00 | 23.00 | 29.18 |
| | baseline | 37.00 | 41.00 | 49.00 | 60.00 | 69.00 | 67.00 | 60.00 | 59.00 | 49.00 | 30.00 | 32.00 | 50.27 |

**Note:**
Results show that excluding any single module during testing changes performance only slightly, whereas excluding both modules degrades performance significantly.

removed, a significant decrease of accuracy appears. Maybe because the number of subjects in CASIA-B is much smaller than OUMVLP, so the model faces less challenges when modules are removed.

For GaitVViT, an ablation study is conducted excluding BNNeck or baseline backbone (*Luo et al., 2020*). 'no BN' means the original BNNeck is replaced by Layer Normalization, 'ResNet' means the original backbone adopted from GaitGL is replaced by a four layers ResNet backbone (*Lin et al., 2022*). Results are shown in Table 5. The data are obtained using single-view gallery evaluation. The results show that the introducing of BNNeck will indeed increase the accuracy in normal walking status, but it appears slightly sensitive to appearance noises as probe sequence changing to 'BG' or 'CL'. Maybe the reason is that in training, batch normalization is carried out when the batches are a mixture of three statuses, but it is not in evaluation, so the features are shifted. The results also show the contribution of the original backbone in generating fine-grained global and local features.

**Table 5 GaitVViT's Rank-1 accuracy on CASIA-B divided in 11 probe view angle and 3 walking status, excluding identical-view cases.**

| Status | Method | Probe view | | | | | | | | | | | Mean |
|---|---|---|---|---|---|---|---|---|---|---|---|---|---|
| | | 0° | 18° | 36° | 54° | 72° | 90° | 108° | 126° | 144° | 162° | 180° | |
| NM | no BN | 95.30 | 99.00 | 99.40 | 97.90 | 94.80 | 93.20 | 96.50 | 99.50 | 99.00 | 98.20 | 93.80 | 96.96 |
| | ResNet | 91.60 | 98.40 | 99.70 | 98.30 | 93.90 | 91.90 | 95.60 | 98.30 | 98.60 | 97.10 | 91.70 | 95.92 |
| | baseline | 96.50 | 99.20 | 99.40 | 98.20 | 96.90 | 94.00 | 96.70 | 99.40 | 99.30 | 98.30 | 93.60 | 97.41 |
| BG | no BN | 90.50 | 95.90 | 95.20 | 92.63 | 90.80 | 82.60 | 89.50 | 95.30 | 96.30 | 95.76 | 88.90 | 92.13 |
| | ResNet | 89.00 | 95.70 | 95.60 | 93.94 | 88.20 | 82.00 | 87.00 | 94.50 | 95.20 | 94.45 | 85.40 | 91.00 |
| | baseline | 90.50 | 95.60 | 95.90 | 93.64 | 89.30 | 82.40 | 88.20 | 94.30 | 96.30 | 94.04 | 90.80 | 91.91 |
| CL | no BN | 67.10 | 85.40 | 87.40 | 84.10 | 77.50 | 73.80 | 78.10 | 81.70 | 86.10 | 83.30 | 68.40 | 79.35 |
| | ResNet | 62.80 | 79.00 | 81.00 | 79.80 | 75.40 | 70.90 | 74.40 | 74.30 | 76.80 | 73.00 | 57.50 | 73.17 |
| | baseline | 67.20 | 81.70 | 86.20 | 82.30 | 76.90 | 70.50 | 75.30 | 80.50 | 84.30 | 80.20 | 62.50 | 77.05 |

Note:
The data were obtained using single-view-gallery evaluation.

**Table 6 GaitTriViT's Rank-1 accuracy *via* different selection methods on CASIA-B in 11 probe view angle and 3 walking status, excluding identical-view cases.**

| Status | Method | Probe view | | | | | | | | | | | Mean |
|---|---|---|---|---|---|---|---|---|---|---|---|---|---|
| | | 0° | 18° | 36° | 54° | 72° | 90° | 108° | 126° | 144° | 162° | 180° | |
| NM | intell 4 | 98.34 | 98.28 | 98.45 | 97.70 | 98.39 | 98.92 | 99.27 | 99.34 | 98.72 | 96.90 | 97.00 | 98.30 |
| | intell full | 100.0 | 100.0 | 100.0 | 100.0 | 100.0 | 100.0 | 100.0 | 100.0 | 100.0 | 100.0 | 99.00 | 99.91 |
| | dense 28 | 97.74 | 96.67 | 95.81 | 95.48 | 97.94 | 96.98 | 97.30 | 97.59 | 96.21 | 95.31 | 96.24 | 96.66 |
| | baseline | 100.0 | 100.0 | 100.0 | 100.0 | 100.0 | 100.0 | 100.0 | 100.0 | 100.0 | 100.0 | 99.00 | 99.91 |
| BG | intell 4 | 89.43 | 85.41 | 89.45 | 87.03 | 92.02 | 87.55 | 89.14 | 89.84 | 87.14 | 82.85 | 81.39 | 87.39 |
| | intell full | 95.00 | 89.00 | 94.00 | 93.94 | 97.00 | 93.00 | 93.00 | 97.00 | 94.00 | 89.90 | 86.00 | 92.89 |
| | dense 28 | 87.43 | 83.21 | 87.37 | 88.49 | 92.20 | 84.88 | 85.87 | 88.89 | 87.72 | 82.62 | 82.74 | 86.49 |
| | baseline | 95.00 | 89.00 | 94.00 | 93.94 | 97.00 | 93.00 | 93.00 | 97.00 | 94.00 | 89.90 | 86.00 | 92.89 |
| CL | intell 4 | 34.13 | 32.48 | 40.16 | 48.26 | 60.36 | 61.73 | 55.39 | 51.38 | 43.15 | 29.99 | 29.92 | 44.27 |
| | intell full | 37.00 | 41.00 | 49.00 | 60.00 | 69.00 | 67.00 | 60.00 | 59.00 | 49.00 | 30.00 | 32.00 | 50.27 |
| | dense 28 | 33.07 | 35.31 | 41.69 | 51.36 | 59.72 | 60.08 | 52.67 | 48.16 | 44.88 | 32.04 | 28.78 | 44.34 |
| | baseline | 37.00 | 41.00 | 49.00 | 60.00 | 69.00 | 67.00 | 60.00 | 59.00 | 49.00 | 30.00 | 32.00 | 50.27 |

Note:
Results show the need of enough frame amount.

### Analysis of different selection methods

In the section state-of-the-art comparison, the frame select strategy of GaitTriViT in the test phase is picking the frames in query sequences serially, *i.e.*, in the order in the frames are shot. It is different from the selection strategy when training, where we pick the required number of frames from the corresponding sub-sequences by cutting the whole sequence. One reason is obvious, the test selection strategy used is more likely to the real-world scenarios, we obtain the silhouettes from the subject sequentially and we can process the task in real-time. The selection strategy in training is named 'intelligent' and the other is named 'dense'. The question is, if the selection strategy in evaluation was the

**Table 7 GaitTriViT's Rank-1 accuracy with or without part embedding on CASIA-B in 11 view angle and 3 walking status, excluding identical-view cases. Accuracy are improved in 'CL' status but dropped in 'BG'.**

| Status | Method | Probe view | | | | | | | | | | | Mean |
|--------|--------|------|------|------|------|------|------|------|------|------|------|------|------|
| | | 0° | 18° | 36° | 54° | 72° | 90° | 108° | 126° | 144° | 162° | 180° | |
| NM | part emb | 100.0 | 100.0 | 100.0 | 100.0 | 100.0 | 100.0 | 100.0 | 100.0 | 100.0 | 99.00 | 99.00 | 99.82 |
| | baseline | 100.0 | 100.0 | 100.0 | 100.0 | 100.0 | 100.0 | 100.0 | 100.0 | 100.0 | 100.0 | 99.00 | 99.91 |
| BG | part emb | 90.00 | 86.00 | 89.00 | 88.89 | 96.00 | 88.00 | 89.00 | 96.00 | 93.00 | 90.91 | 82.00 | 89.89 |
| | baseline | 95.00 | 89.00 | 94.00 | 93.94 | 97.00 | 93.00 | 93.00 | 97.00 | 94.00 | 89.90 | 86.00 | 92.89 |
| CL | part emb | 37.00 | 43.00 | 55.00 | 62.00 | 69.00 | 65.00 | 62.00 | 53.00 | 44.00 | 35.00 | 32.00 | 50.64 |
| | baseline | 37.00 | 41.00 | 49.00 | 60.00 | 69.00 | 67.00 | 60.00 | 59.00 | 49.00 | 30.00 | 32.00 | 50.27 |

**Note:**

Results indicate that incorporating part embedding marginally improves GaitTriViT's accuracy for subjects wearing coats, whereas it degrades performance when subjects carry bags.

same as training, will the model perform better or not. Several tests with different frame selection methods are conducted and the results are shown in Table 6.

In Table 6, the 'intell 4' means 4 frames are selected through 'intelligent' strategy (*i.e.*, same way when training) and run the inference independently; 'intell full' uses all probe sequence frames to conduct the inference; 'dense 28' using 28 frames for inference while not changing the original test selection strategy. The results show that in inference, more frames selected means higher evaluation accuracy. But when frames amount is small, using 'intelligent' strategy will improve the performance.

### Analysis of part embeddings

Given the inspiration of position embedding in original Vision Transformer and implementation of angle embedding and case embedding in GaitTriViT (*Dosovitskiy et al., 2020*), these additional learnable embedding shows its value. Prior works also indicate the effectiveness of the lightweight learnable embedding for learning invariant non-visual features (*He et al., 2021*; *Peng et al., 2023*). So, it may also help non-context manual intervention like feature map partition and improve the model performance. The parameters are initialized with pre-trained GaitTriViT baseline checkpoint and fine-tuned with 80 epochs. The results are shown in Table 7.

The results show that adding the part embedding slightly increases the accuracy of GaitTriViT when subjects wear coats. Because in proposed GaitTriViT, the feature map is divided into four strips, which may roughly correspond to head and chest, waist and arms, crotch and thigh, as well as lower legs and feet. So, the part embedding will learn which part to emphasize. For gait sequences wearing a coat, the top three body parts are all self-occluded or blurred, so less discriminative representations can be extracted from the feature maps. Thus, the model will tend to focus more attention on the bottom part which only has lower legs and feet. For wearing coat status, this change is beneficial, but for the normal walking probe and bag carrying probe, this change causes less attention on their information-riches top three parts. So, they may face an accurate drop.

**Table 8 GaitTriViT's rank-1 accuracy with different TCSS order on CASIA-B in 11 view angle and 3 walking status, excluding identical-view cases.**

| Status | Method | Probe view | | | | | | | | | | | Mean |
|--------|--------|------|------|------|------|------|------|------|------|------|------|------|------|
| | | 0° | 18° | 36° | 54° | 72° | 90° | 108° | 126° | 144° | 162° | 180° | |
| NM | TCSS after | 100.0 | 100.0 | 100.0 | 100.0 | 100.0 | 100.0 | 100.0 | 100.0 | 100.0 | 100.0 | 100.0 | 100.0 |
| | baseline | 100.0 | 100.0 | 100.0 | 100.0 | 100.0 | 100.0 | 100.0 | 100.0 | 100.0 | 100.0 | 99.00 | 99.91 |
| BG | TCSS after | 87.00 | 82.00 | 88.00 | 90.91 | 95.00 | 85.00 | 82.00 | 92.00 | 85.00 | 84.85 | 80.00 | 86.52 |
| | baseline | 95.00 | 89.00 | 94.00 | 93.94 | 97.00 | 93.00 | 93.00 | 97.00 | 94.00 | 89.90 | 86.00 | 92.89 |
| CL | TCSS after | 33.00 | 34.00 | 44.00 | 48.00 | 65.00 | 65.00 | 62.00 | 52.00 | 43.00 | 32.00 | 28.00 | 46.00 |
| | baseline | 37.00 | 41.00 | 49.00 | 60.00 | 69.00 | 67.00 | 60.00 | 59.00 | 49.00 | 30.00 | 32.00 | 50.27 |

**Note:**
Results indicate the order has minimal impact. Altering the order fails to boost feature discriminability while eroding robustness.

### Analysis of order between shuffle and partition

In the GaitTriViT baseline, the TCSS with partition operation raises a question about the correct order of two operations. The part-dependent idea is like a manual operation to tell the model where it belongs to an independent region that has different features from other regions, *i.e.*, different morphological characteristics between the limbs. Therefore, the original order where shift and shuffle are previous than partition will first shuffle within the whole feature map, which may make the partition meaningless. Therefore, this section tries to move the TCSS module after the partition, wondering if the different order of TCSS and partition operations will make some changes. The model with different modules order is re-trained for 50 epochs. The results and comparison are shown in Table 8.

The results in Table 8 show almost no improvement with the employment of TCSS after strategy. At every probe view of 'BG' status and almost every probe view of 'CL', the results encounter a certain percentage of decline. Results show the order change will not increase the feature discriminativeness but lose its robustness.

## CONCLUSIONS

This article proposes two customized Transformer-based gait recognition models, GaitTriViT and GaitVViT, to extract fine-grained features representing human walking patterns. For GaitTriViT, this work utilizes the rapidly evolving Vision Transformer instead of traditional convolutional neural networks to build the model, in contrast to the gait recognition pipeline, a strategy is employed that makes the temporal pooling (TP) module and horizontal pooling (HP) module in parallel. By incorporating Vision Transformer and Spatio-temporal Attention mechanism, the temporal-global features are obtained in global temporal branch. The model also utilizes part-dependent and shuffle strategies to extract spatial-local features in the local spatial branch, resulting in fine-grained features without down-sampling. For GaitVViT, dissatisfied with the design of the TP module in the gait recognition common framework, the Video Vision Transformer is introduced for enhancement. The proposed Video ViT Encoder will take the output of GaitVViT backbone as sequence of patches. Thus, encoder extracts the spatial feature at temporal dimension and generates the final spatio-temporal feature.

Evaluation results demonstrate that the proposed method GaitTriViT meets quite a challenge on both the popular benchmarks: CASIA-B and OUMVLP, while the other proposed method GaitVViT reaches the line of state-of-the-art models based on traditional convolutional neural networks. Evaluations compare between two proposed methods and argue that current vision transformer structure may not be a good upstream backbone for binary inputs like gait silhouette. Modification and improvement are compulsory to tackle this challenge. The author still believes in the potential of Transformer-based structure in gait recognition as well as other video-based recognition tasks.

## Limitations

In this work, two Transformer-based methods GaitTriViT and GaitVViT are proposed to tackle the gait recognition task. On two popular benchmarks: CASIA-B and OUMVLP, GaitTriViT meets huge difficulties, the results only surpass the GEINet method leveraging gait energy images (GEI) for temporal modeling. Among its own results, GaitTriViT also has a lot of limitations. On the smaller CASIA-B dataset, based on different walking status, there are three scenarios: normal walking, bag carrying and wearing coat. Compared to probe status of normal walking, the evaluation in bag carrying status encounters a reasonable drop-down relatively. However, in the case of subjects wearing coats, there was a significant performance drop during evaluation, highlighting a lack of robustness in our method when silhouette appearances have significant changes. For GaitVViT, although the performance has met the acceptable level on both popular benchmarks. It's still a little far away from cutting-edge methods. It is not enough for a temporal augmented method. And the sensitivity toward appearance noise also needs addressing. Furthermore, Vision Transformer also has its inherent limitations, *e.g.*, computation consuming and demand for large datasets, all need extra works in future.

## Future works

Given the lack of robustness in GaitTriViT when silhouette appearances have significant changes. If given the opportunity to work on this further, work will be focused on the robustness to minimize the noises of appearance, the influence of subjects' walking frequency on gait patterns needs to be addressed too. For example, inserting a module to separate the appearance and gait features. For GaitVViT, the generalization needs no worry, it is encouraging to combine the tricks of GaitTriViT (*e.g.*, angel embedding) and the structure of GaitVViT together, the fusion of two methods may help to pursue better performance. Technologies like data augmentation and reinforcement learning may also contribute to the task.

For the inherent limitations of Vision Transformer itself, we might introduce the idea of latent and running the model in latent space, leveraging powerful encoders, *e.g.*, variational auto-encoder (VAE) and denoising diffusion implicit model (DDIM). They may help the model to capture high-level patterns better and increase robustness toward noise as well as efficiency.

Moreover, real-world applications are challenging, rather than popular in-door silhouette datasets, this work needs to face the in-wild datasets captured from real-world

scenes, integrating target detection and image segmentation modules. The target would be to enhance the model's robustness when facing variations in subject appearances, and ultimately, testing a capable real-world application to evaluate the capabilities under diverse conditions.

### Funding
The authors received no funding for this work.

### Competing Interests
The authors declare that they have no competing interests.

### Author Contributions
• Hongyun Sheng conceived and designed the experiments, performed the experiments, analyzed the data, performed the computation work, prepared figures and/or tables, authored or reviewed drafts of the article, and approved the final draft.

### Data Availability
The code is available in the Supplemental File.

The CASIA Gait data is available at: http://www.cbsr.ia.ac.cn/english/Gait%20Databases.asp.

The OU-ISIR Biometric Database is available at: http://www.am.sanken.osaka-u.ac.jp/BiometricDB/GaitMVLP.html.

### Supplemental Information
Supplemental information for this article can be found online at http://dx.doi.org/10.7717/peerj-cs.3061#supplemental-information.

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
