# Peer review of "GaitTriViT and GaitVViT: Transformer-based methods emphasizing spatial or temporal aspects in gait recognition"

_PeerJ Computer Science, doi:10.7717/peerj-cs.3061_

## Round 0.1 · original submission · Major Revisions

Dear Authors,

The reviews for your manuscript are included at the bottom of this letter. We ask that you make changes to your manuscript based on those comments. Reviewers have asked you to provide specific references. You are welcome to add them if you think they are useful and relevant. However, you are not obliged to include these citations, and if you do not, it will not affect my decision.

Furthermore, some of the figures have very low resolution and they should be polished.

Explanation of the equations should be checked. All variables should be written in italics as in the equations. Their definitions and boundaries should be defined. Necessary references should also be given. Many of the equations are part of the related sentences. Attention is needed for correct sentence formation.

Too much first-person pronouns should be avoided. "I chose", "I argue", "I still believe", "I argued", "I discuss", "I compare", "I carried", " I also carry", "I set pairs of", "I conduct a", "I still believe", "I will focus", "I may insert", " I could combine", and etc should be corrected. Future Works should be completely rewritten.

"Chapter" should be changed as "Section" in the manuscript.

Some citations do not seem in the References sections. "Tan & Tan, 2006" is absent. All in-text references should be listed in the reference list.

Best wishes,

Reviewer 1 ·

Basic reporting

The paper is relatively well written. The introduction section properly defines the terms used in the paper, as well as overall trends in this area, motivating the proposed approach.

Since this paper is mainly about the use of a "custom" vision transformer encoder architecture for gait recognition, there are many other concurrent works that proposed such variants in literature, which I expect the author to cite and discuss in the related work section. Furthermore, this calls into question the claim that the work presented is "novel". Even though the proposed architecture is carefully designed and achieves good results, I would not use this word when describing the contributions.

@inproceedings{catruna24gaitpt,
author = {Andy Catruna and Adrian Cosma and Emilian Radoi},
title = {GaitPT: Skeletons are All You Need for Gait Recognition},
booktitle = {18th {IEEE} International Conference on Automatic Face and Gesture Recognition, {FG} 2024, Istanbul, Turkey, May 27-31, 2024},
pages = {1--10},
publisher = {{IEEE}},
year = {2024},
url = {https://doi.org/10.1109/FG59268.2024.10581947},
doi = {10.1109/FG59268.2024.10581947},
timestamp = {Wed, 31 Jul 2024 14:28:05 +0200},
biburl = {https://dblp.org/rec/conf/fgr/CatrunaCR24b.bib},
bibsource = {dblp computer science bibliography, https://dblp.org}
}

@Article{cosma23gaitvit,
AUTHOR = {Cosma, Adrian and Catruna, Andy and Radoi, Emilian},
TITLE = {Exploring Self-Supervised Vision Transformers for Gait Recognition in the Wild},
JOURNAL = {Sensors},
VOLUME = {23},
YEAR = {2023},
NUMBER = {5},
ARTICLE-NUMBER = {2680},
URL = {https://www.mdpi.com/1424-8220/23/5/2680},
ISSN = {1424-8220},
DOI = {10.3390/s23052680}
}

@Article{cosma20wildgait,
AUTHOR = {Cosma, Adrian and Radoi, Ion Emilian},
TITLE = {WildGait: Learning Gait Representations from Raw Surveillance Streams},
JOURNAL = {Sensors},
VOLUME = {21},
YEAR = {2021},
NUMBER = {24},
ARTICLE-NUMBER = {8387},
URL = {https://www.mdpi.com/1424-8220/21/24/8387},
PubMedID = {34960479},
ISSN = {1424-8220},
DOI = {10.3390/s21248387}
}

@Article{cosma22gaitformer,
AUTHOR = {Cosma, Adrian and Radoi, Emilian},
TITLE = {Learning Gait Representations with Noisy Multi-Task Learning},
JOURNAL = {Sensors},
VOLUME = {22},
YEAR = {2022},
NUMBER = {18},
ARTICLE-NUMBER = {6803},
URL = {https://www.mdpi.com/1424-8220/22/18/6803},
ISSN = {1424-8220},
DOI = {10.3390/s22186803}
}

There are some formatting issues, although I am not sure if it is because of the PeerJ template or not. For example, in the explanation for Equation (1) (line 298), the symbols are not properly rendered (they appear as white squares). That also happens at line 322, 407, etc. Again, not sure if it is because of the PeerJ template, but the figures are missing in the main text and appear very stretched only at the end of the PDF.

Section 3.4.2 "backbone" (line 420) needs capitalization.

Experimental design

This experimental setup is widely used in gait recognition literature. However, the author only evaluated on controlled gait recognition scenarios (CASIA-B / OU-ISIR), which in my opinion, already have their performance saturated. The paper would be better if the author included some in-the-wild scenarios, like the GREW dataset and some others.

The authors properly included hyperparameters, making the reproduction of the work easy.

Validity of the findings

No comment - the author developed a vision transformer-based architecture for processing gait silhouettes and obtained slightly better results than the previous work.

Additional comments

-

Reviewer 2 ·

Basic reporting

The abstract includes excessive technical details, such as benchmarks and metrics, which would be better suited in the results section.
While the methods (GaitTriViT and GaitVViT) are described, the explanation of what makes these approaches fundamentally unique compared to state-of-the-art methods is vague.
The inclusion of certain modules, such as Temporal Clips Shift and Shuffle (TCSS), lacks justification in terms of how they contribute to performance improvements.
Figures, such as those depicting the pipeline and architecture, are referenced but not fully explained in the text. This leaves readers unclear about their specific relevance.
The paper briefly mentions the datasets (CASIA-B and OUMVLP) but lacks sufficient details on their preprocessing, such as image alignment and cropping.

Experimental design

Provide more details about preprocessing, dataset splitting, and parameter selection.

While the manuscript compares its methods to existing ones, there is no clear rationale for choosing specific baselines, such as GaitGL or GaitSet, and why others were excluded.
The evaluation methodology is described (e.g., single-view-gallery and cross-view-gallery evaluation), but the rationale for these choices and their implications on real-world applications are not discussed in depth.
While ablation studies are conducted, the findings are not discussed in sufficient detail, particularly regarding their implications for model design and performance.
While the manuscript mentions using Kaiming initialization and ImageNet pretraining, it does not explore alternative strategies or justify their effectiveness for gait recognition.
The combination of loss functions (e.g., triplet loss, cross-entropy loss, and attention loss) is described but not evaluated in isolation to demonstrate their individual contributions.
While ViTs and Video ViTs are innovative, the paper does not adequately discuss their inherent limitations, such as computational overhead or potential issues with overfitting due to large parameter spaces.

Validity of the findings

Metrics such as rank-1 accuracy are reported, but their real-world implications for gait recognition tasks are not analyzed sufficiently.
The results suggest that the proposed methods, particularly GaitTriViT, are highly sensitive to appearance noise (e.g., subjects carrying bags or wearing coats). This limitation is acknowledged but not addressed effectively.
GaitTriViT fails to significantly outperform state-of-the-art methods in many scenarios. While GaitVViT shows improvements, its performance gain is marginal and not aligned with the claims of transformative innovation.

Compare the results with the recent approaches like:
https://doi.org/10.1007/s13198-024-02508-3
https://doi.org/10.1016/j.imavis.2023.104721
Address sensitivity to appearance noise with potential solutions, such as additional data augmentation or adaptive feature extraction techniques.
Improve figures and tables with clear captions and thorough explanations in the text.
Acknowledge the limitations of transformer-based approaches and discuss plans for future improvements.

Additional comments

Numerous grammatical errors and awkward phrasing reduce readability. For instance, the abstract has overly complex sentences, which could be simplified for clarity.
Concepts such as the advantages of Vision Transformers (ViTs) over CNNs and their application in gait recognition are repeated unnecessarily in multiple sections.
The paper’s structure is not streamlined. For instance, the related works section could be more concise, focusing only on directly relevant studies.
Concepts such as the benefits of spatial and temporal feature extraction are reiterated in multiple sections unnecessarily.
Revise for grammar, sentence flow, and conciseness, particularly in the abstract and introduction.
Clearly articulate the novelty of the proposed methods compared to existing approaches, with specific examples.

---

## Round 0.2 · Minor Revisions

Dear Authors,

One of the previous reviewers, who requested minor revision, did not accept the intivation for reviewing the revised manuscript. Other reviewer still thinks that the paper needs revision. Please address the concerns of this reviewer. By the way the reviewe has recommended that specific references be provided. Should you deem them relevant and useful, you are at liberty to add them. However, inclusion of these is not obligatory, and the absence of such content will not influence the decision-making process.

Best wishes,

Reviewer 2 ·

Basic reporting

The author need to provide a mathematical model regarding how the frame is processed within the proposed network for better clarity and reproducibility of the work.

Experimental design

Point 8: While ablation studies are conducted, the findings are not discussed in sufficient detail, particularly regarding their implications for model design and performance; While the manuscript mentions using Kaiming initialization and ImageNet pretraining, it does not explore alternative strategies or justify their effectiveness for gait recognition; The combination of loss functions (e.g., triplet loss, cross-entropy loss, and attention loss) is described but not evaluated in isolation to demonstrate their individual contributions.

Response 8: Thank you for your insightful feedback. We appreciate the opportunity to clarify these important methodological considerations. While we fully agree with the scientific value of the suggested analyses, current resource constraints (particularly in GPU allocation and computational budget) prevent us from conducting additional large-scale experiments within the revision timeframe. In future work, we will fully consider these experiments to comprehensively verify the performance of the algorithm

The authors need to conduct the ablation analysis for proper justification of the designed network.

Validity of the findings

I would like to reiterate the question:
Compare the results with the recent approaches like:
https://doi.org/10.1007/s13198-024-02508-3
https://doi.org/10.1016/j.imavis.2023.104721

---

## Round 0.3 · accepted · Accept

Dear Authors,

Thank you for clearly addressing the reviewer's comments.

Best wishes,

Reviewer 2 ·

Basic reporting

The manuscript has been substantially improved.

Experimental design

No further comments.

Validity of the findings

No further comments.